# *Fgf4* maintains *Hes7* levels critical for normal somite segmentation clock function

**Matthew J Anderson[1], Valentin Magidson[2], Ryoichiro Kageyama[3], Mark Lewandoski[1]***

[1]Genetics of Vertebrate Development Section, Cancer and Developmental Biology Laboratory, National Cancer Institute, National Institutes of Health, Frederick, United States; [2]Optical Microscopy and Analysis Laboratory, Frederick National Laboratory for Cancer Research, Frederick, United States; [3]Institute for Frontier Life and Medical Sciences, Kyoto University, Kyoto, Japan

**Abstract** During vertebrate development, the presomitic mesoderm (PSM) periodically segments into somites, which will form the segmented vertebral column and associated muscle, connective tissue, and dermis. The periodicity of somitogenesis is regulated by a segmentation clock of oscillating Notch activity. Here, we examined mouse mutants lacking only *Fgf4* or *Fgf8*, which we previously demonstrated act redundantly to prevent PSM differentiation. *Fgf8* is not required for somitogenesis, but *Fgf4* mutants display a range of vertebral defects. We analyzed *Fgf4* mutants by quantifying mRNAs fluorescently labeled by hybridization chain reaction within Imaris-based volumetric tissue subsets. These data indicate that FGF4 maintains *Hes7* levels and normal oscillatory patterns. To support our hypothesis that FGF4 regulates somitogenesis through *Hes7*, we demonstrate genetic synergy between *Hes7* and *Fgf4*, but not with *Fgf8*. Our data indicate that *Fgf4* is potentially important in a spectrum of human Segmentation Defects of the Vertebrae caused by defective Notch oscillations.

*For correspondence:
lewandom@mail.nih.gov

Competing interests: The authors declare that no competing interests exist.

## Introduction

A common developmental mode employed by many embryos is a segmentation clock that oscillates within a posterior growth zone; with each cycle, a segment forms. This stratagem has evolved independently within the three major bilaterian clades (annelids, arthropods, and chordates) as well as in plants (*Clark et al., 2019*; *Richmond and Oates, 2012*; *Chipman, 2010*; *Ten Tusscher, 2020*; *Balavoine, 2014*). In chordates, a segmentation clock oscillates in the presomitic mesoderm (PSM); with each cycle, a pair of somites form flanking the neural tube (*Pourquié, 2011*; *Hubaud and Pourquié, 2014*). Somites differentiate into dermis, skeletal muscle, tendons, as well as the vertebral column, which retains the segmented attribute of the somites (*Christ et al., 2007*).

In vertebrates, genes with an oscillatory pattern within the PSM include those encoding components or targets of the FGF, WNT, and Notch signaling pathways (*Hubaud and Pourquié, 2014*; *Krol et al., 2011*; *Dequéant et al., 2006*). However, which individual genes oscillate differs among species, with the exception of the Notch-responsive HES/HER transcription factors, suggesting that Notch signaling is at the core of the somitogenesis clock (*Krol et al., 2011*). Supporting this idea, genetic and pharmacological manipulations demonstrate that Notch pathway oscillation in the mouse embryo is essential for somitogenesis (*Ferjentsik et al., 2009*). In the mouse, oscillatory waves of the *Notch1* receptor and Delta-like 1 (*Dll1*) ligand expression, as well as the activated Notch receptor (cleaved Notch intracellular domain, NICD) sweep from the posterior to anterior, where oscillations arrest (*Bone et al., 2014*; *Shimojo et al., 2016*). These oscillations are established

through negative-feedback loops of Notch components such as the transcriptional repressor, HES7 (*Niwa et al., 2011*), and the glycosyltransferase, LFNG (*Morimoto et al., 2005*), both of which are encoded by oscillating genes under Notch pathway regulation (*Niwa et al., 2011*). Notch oscillations arrest in the anterior PSM, where NICD cooperates with TBX6 to periodically activate *Mesp2* expression (*Oginuma et al., 2008*). MESP2 then regulates formation of the nascent somite as well as its rostro-caudal patterning, which is essential for normal patterning of the subsequent vertebral column (*Takahashi et al., 2000*; *Saga et al., 1997*).

Mutations in nearly all of the aforementioned Notch pathway genes have been identified in human patients where defective somitogenesis is thought to be the cause of Segmentation Defects of the Vertebrae (SDV) (*Eckalbar et al., 2012*; *Giampietro et al., 2009*). For example, frequently recessive mutations in *DLL3*, *HES7*, *MESP2* and *LFNG* and a dominant mutation in *TBX6* (*Sparrow et al., 2013*; *Lefebvre et al., 2017*) have been identified in patients with spondylocostal dysostosis (SCDO), which is characterized by severe vertebral malformations that include hemivertebrae, vertebral loss and fusion along the length of the axis (*Takeda et al., 2018*). Whereas SCDO is relatively rare, congenital scoliosis (CS), defined as a lateral curvature of the spine exceeding 10%, is much more common, with a frequency of 1:1000, which is suspected to be an underestimation because asymptomatic individuals do not seek medical care (*Giampietro et al., 2013*). Mutated alleles of *HES7*, *LFNG*, *MESP2*, and *TBX6* are all associated with CS (*Lefebvre et al., 2017*; *Takeda et al., 2018*; *Giampietro et al., 2013*; *Sparrow et al., 2012*).

The position in the anterior PSM where Notch oscillations arrest and *Mesp2* expression establishes the future somite boundary is called the determination front (*Morimoto et al., 2005*; *Saga et al., 1997*; *Dubrulle et al., 2001*). This region is also the anterior limit of the wavefront in the classical clock-and-wavefront, a theoretical model proposed over 40 years ago to explain the precipitous and periodic formation of segments in the PSM (*Cooke and Zeeman, 1976*). In this model, wavefront activity prevents the PSM from responding to the segmentation clock; hence somites form only anteriorly, where the wavefront ends. In chick or zebrafish embryos, exogenously added FGF protein or pharmacological inhibition of FGF signaling will shift the determination front rostrally or caudally, respectively. In the mouse, Cre-mediated inactivation of *Fgf4* and *Fgf8* specifically in the PSM results in an initial expansion of *Mesp2* expression, followed by the premature expression of somite markers throughout the PSM (*Naiche et al., 2011*). Canonical WNT signaling is also a wavefront candidate, mostly because ectopic activation of βcatenin in the PSM results in an expansion of this tissue and a rostral shift of *Mesp2* expression (*Aulehla et al., 2008*; *Dunty et al., 2008*). However, this expansion observed in βcatenin 'gain-of-function' mutants requires FGF4 and FGF8 activity, suggesting that these FGF signals are synonymous with the wavefront (*Naiche et al., 2011*).

In addition to its role in preventing somite differentiation, FGF signaling is also implicated in the regulation of key Notch components of the segmentation clock. Several studies have demonstrated that pharmacological inhibition of FGF disrupts Notch oscillations (*Wahl et al., 2007*; *Niwa et al., 2007*). In mouse embryos with PSM-specific loss of *FGF receptor one* or both *Fgf4* and *Fgf8*, Notch pathway genes such as *Hes7*, are downregulated, although this analysis can be complicated due to the loss of PSM tissue in these mutants (*Naiche et al., 2011*; *Niwa et al., 2007*). Here, we focus on FGF-Notch interactions by analyzing mouse mutants that lack only one of the wavefront Fgf genes, *Fgf8* or *Fgf4*. An indispensable technique in our analysis is whole mount in situ hybridization chain reaction (HCR), which allows us to multiplex different gene expression domains and quantify mRNA levels within specific embryonic tissues (*Choi et al., 2014*; *Choi et al., 2018*; *Trivedi et al., 2018*). We demonstrate that, while *Fgf8* is not required for somitogenesis, *Fgf4* is required for normal Notch oscillatory patterns and patterning of the vertebral column.

## Results

### FGF4 activity is required in the PSM for normal segmentation of rostral somites

To analyze the role of *Fgf4* or *Fgf8* expression in the primitive streak and PSM (*Figure 1A* - B), we inactivated each gene specifically within these tissues using the transgenic mouse line, Tg(T-cre) 1Lwd (hereafter 'TCre'), in which Cre is controlled by *Brachyury* regulatory elements (*Perantoni et al., 2005*). Thus, we generated '*Fgf4* mutants' (TCre; *Fgf4*<sup>flox/Δ</sup>; 'Δ' = 'deleted' or null)

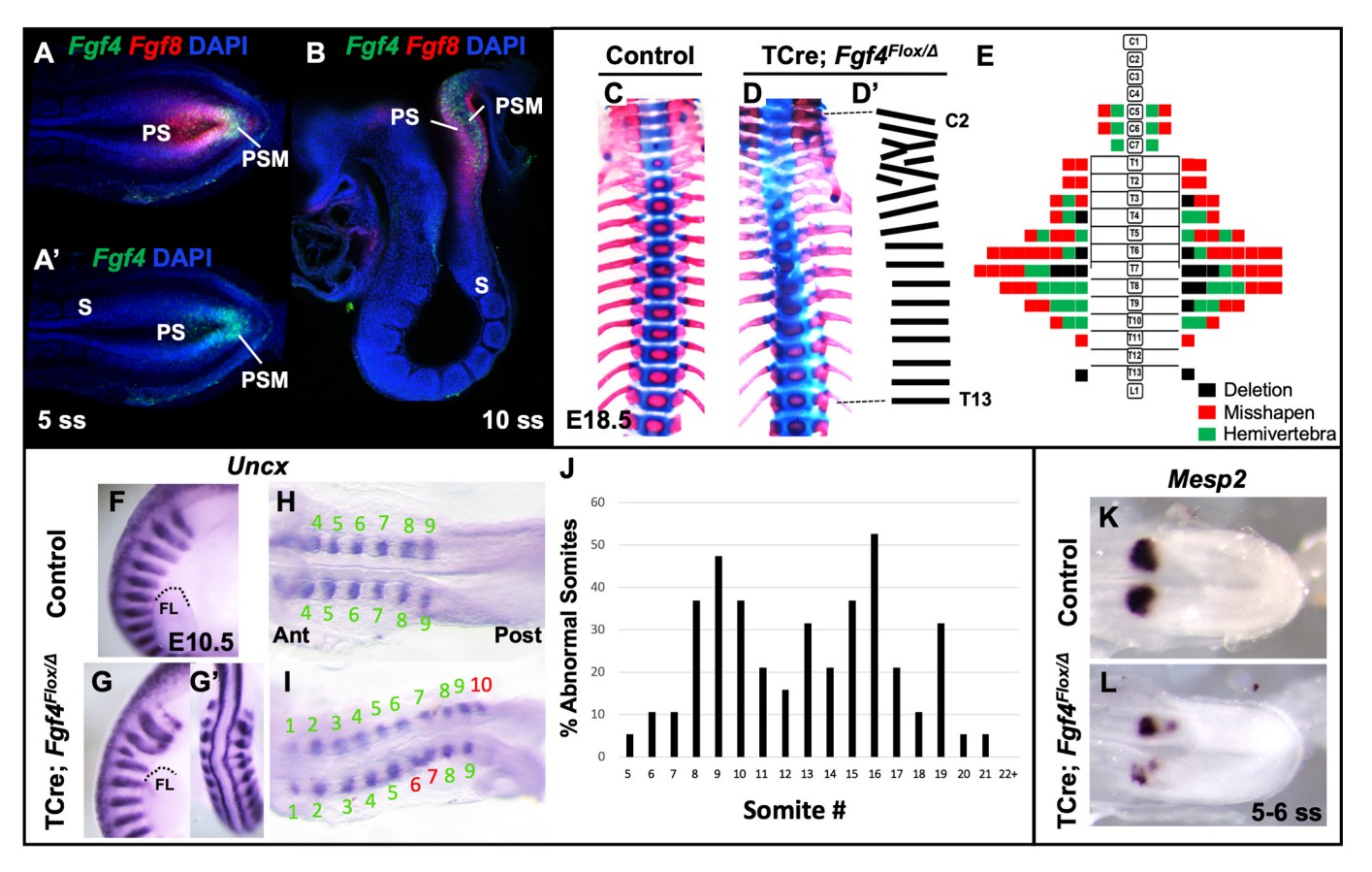

**Figure 1.** *Fgf4* mutants have defects in vertebral and somite patterning. (A, A') Max intensity projection (MIP) of HCR staining of *Fgf4* and *Fgf8* mRNA expression (A) or *Fgf4* only (A') in the PSM of a five somite stage wildtype embryo (dorsal view). PS: primitive streak, PSM: presomitic mesoderm, S: somite. (B) MIP of 4 parasagittal z-sections (approximately 5 μm each) of HCR staining of *Fgf4* and *Fgf8* mRNA expression, of a 10 somite stage wildtype embryo (lateral view, mediolateral to the midline). (C, D) Skeletal preparations (ventral view after removing ribcage) of E18.5 control and *Fgf4* mutants. (D') Tracing of the *Fgf4* mutant vertebral pattern in D. (E) Histograms representing the vertebral column and associated ribs (C = cervical, T = thoracic, L = lumbar), showing variety and location of vertebral defects in E18.5 *Fgf4* mutants (n = 12). Each block in the histogram represents a single defect as indicated in the key, bottom right. (F–I) Wholemount in situ hybridization (WISH) detection of *Uncx* mRNA expression in control and *Fgf4* mutants at E10.5 (F, G lateral views, anterior top; G' dorsal view of embryo in G) and at 9–10 somite stage (H, I dorsal view, anterior left); dotted line in F and G marks the anterior boundary of the forelimb (FL). (J) Graph depicting the percentage of abnormal somites scored with a mispatterned *Uncx* mRNA WISH pattern (F–I) in *Fgf4* mutants at the somite position listed on the x-axis. Somites were scored in 26–30 somite stage *Fgf4* mutants (n = 19). Note that posterior to somite 21 the *Uncx* pattern is normal. (K, L) Dorsal view (anterior left) of 5–6 somite stage control (K) and *Fgf4* mutant (L) embryos WISH-stained for *Mesp2* mRNA expression. In all control embryos (n = 7), the *Mesp2* pattern was normal; in 5/8 mutant embryos, the pattern was abnormal.

The online version of this article includes the following figure supplement(s) for figure 1:

**Figure supplement 1.** Mispatterning of rostral somites correct as development proceeds.

**Figure supplement 2.** *Mesp2* pattern is normal at 24–26 somite stage in *Fgf4* mutants.

**Figure supplement 3.** *Fgf4* expression in mutant and wildtype embryos.

or '*Fgf8* mutants' (TCre; *Fgf8* flox/Δ; see *Table 1*). Whereas *Fgf8* mutants do not survive much beyond birth due to kidney agenesis (*Perantoni et al., 2005*), *Fgf4* mutants are viable and found at Mendelian ratios at weaning (n = 18 controls and n = 21 mutants). Skeleton preparations of *Fgf8* mutant embryos at E18.5 (n = 22) revealed all vertebral bodies were present and normally patterned. *Fgf8* mutants also presented with minor cervical and/or lumbar homeotic transformations, each with incomplete penetrance and expressivity: small ribs were sometimes present in the most posterior cervical vertebra (8/22, 4 bilateral and 4 unilateral) or on the most anterior lumbar vertebra (8/22, 4 bilateral and 4 unilateral). On the other hand, *Fgf4* mutants display a variety of segmentation defects

**Table 1.** Experimental Crosses.

| Experimental cross | Experimental genotype (Freq) | Control genotype (Freq) |
|---|---|---|
| $Fgf4^{flox/flox}$ x $TCre^{TG/TG}$; $Fgf4^{\Delta/wt}$ | $TCre^{TG/0}$; $Fgf4^{flox/\Delta}$ (1/2) | $TCre^{TG/0}$; $Fgf4^{flox/wt}$ (1/2) |
| $Fgf8^{flox/flox}$ x $TCre^{TG/TG}$; $Fgf8^{\Delta/wt}$ | $TCre^{TG/0}$; $Fgf8^{flox/\Delta}$ (1/2) | $TCre^{TG/0}$; $Fgf8^{flox/wt}$ (1/2) |
| $Hes7^{\Delta/wt}$ x $Hes7^{\Delta/wt}$ | $Hes7^{\Delta/\Delta}$ (1/4) | $Hes7^{wt/wt}$ (1/4) |
| $Fgf4^{flox/flox}$; $Hes7^{\Delta/wt}$ x $TCre^{TG/TG}$; $Fgf4^{\Delta/wt}$ | $TCre^{TG/0}$; $Fgf4^{flox/\Delta}$; $Hes7^{\Delta/wt}$ (1/4) | $TCre^{TG/0}$; $Fgf4^{flox/wt}$ (1/4)<br>$TCre^{TG/0}$; $Fgf4^{flox/wt}$; $Hes7^{\Delta/wt}$ (1/4)<br>$TCre^{TG/0}$; $Fgf4^{flox/\Delta}$ (1/4) |
| $Fgf8^{flox/flox}$; $Hes7^{\Delta/wt}$ x $TCre^{TG/TG}$; $Fgf8^{\Delta/wt}$ | $TCre^{TG/0}$; $Fgf8^{flox/\Delta}$; $Hes7^{\Delta/wt}$ (1/4) | $TCre^{TG/0}$; $Fgf8^{flox/wt}$ (1/4)<br>$TCre^{TG/0}$; $Fgf8^{flox/wt}$; $Hes7^{\Delta/wt}$ (1/4)<br>$TCre^{TG/0}$; $Fgf8^{flox/\Delta}$ (1/4) |

Experimental cross is always written 'female x male'.

$\Delta$ = 'deletion' or null.

in the cervical and thoracic vertebrae with 100% penetrance (Compare *Figure 1D,D'* with C). An average of 7.9 defects occurred per mutant and consisted of hemivertebrae, and misshapen and deleted vertebrae (*Figure 1E*).

We examined gross somite patterning in *Fgf4* mutants by staining embryos at various stages for *Uncx* mRNA, which marks the posterior somite compartment (*Mansouri et al., 1997*; *Neidhardt et al., 1997*). Analysis of 19 E9.5 *Fgf4* mutants revealed a frequency of irregular *Uncx* expression specifically in future cervical and thoracic somites with full penetrance (*Figure 1F–J*). We did not score the most rostral somites in these samples because *Uncx* expression is absent in these early somites due to the differentiation of somites 1–4 to form the occipital bones (*Spörle and Schughart, 1997*). To examine these rostral somites we scored 20 *Fgf4* mutants at somite stage seven and found defective *Uncx* patterning beginning at somite three in this sample set (black bars, *Figure 1—figure supplement 1*). Interestingly, a majority of *Uncx* patterning defects in somite 5 and 6 had apparently corrected by somite stage 25 (gray bars, *Figure 1—figure supplement 1*). Hence, these somites partially corrected or regulated their pattern and may account for the relatively low level of rostral vertebral defects at E18.5 (*Figure 1E*). This correction of rostral somite patterning is found in embryos lacking *Hes7* expression (*Bessho, 2001*), which, as we show below, is reduced in *Fgf4* mutants.

To determine if these malformed somites were due to an error in segmentation, we examined *Mesp2* expression, which occurs in the anterior PSM and is required for normal somite segmentation and rostral-caudal somite identity (*Saga et al., 1997*). At the stages when irregular somites are emerging from *Fgf4* mutant PSM, we detected aberrant *Mesp2* expression, whether we analyzed gene expression by traditional wholemount in situ hybridization (WISH) staining (*Figure 1K–L*) or by fluorescent HCR analysis (*Figure 2A,B*). However, at later stages (22–24 somite stages), when properly segmented somites are emerging from *Fgf4* mutant PSM, *Mesp2* expression appears normal (*Figure 1—figure supplement 2*).

We considered whether this normal *Mesp2* expression (*Figure 1—figure supplement 2*) and the resulting proper somite patterning caudal to somite 22 that occurs in *Fgf4* mutants (*Figure 1J*), might reflect an absence of PSM-specific *Fgf4* expression in normal development at later stages. However, we detected *Fgf4* expression throughout the PSM at somite stage 22 through E12.5 (*Figure 1—figure supplement 3C,C', E–G'*). We then considered whether TCre-mediated inactivation of *Fgf4* resulted in loss of *Fgf4* expression in the PSM throughout development. We expected this would be the case because TCre is active in the primitive streak at E7.5 (*Perantoni et al., 2005*) and, in prior studies, has been shown to inactivate *Fgf4* and *Fgf8* in the primitive streak and PSM, prior to formation of the first somite (*Naiche et al., 2011*; *Perantoni et al., 2005*). Indeed, we detect no *Fgf4* expression in the PSM at somite stage 6 (*Figure 1—figure supplement 3B,B'*), when aberrant somites form, nor at somite stage 24 (*Figure 1—figure supplement 3D,D'*), when normal somites form. Therefore, we conclude that *Fgf4* is expressed within the PSM throughout development, but is required for normal somitogenesis only at early stages. When *Fgf4* is inactivated early, in the caudal embryo, abnormally patterned *Mesp2* expression occurs only at these early stages, causing the observed vertebral defects in *Fgf4* mutants.

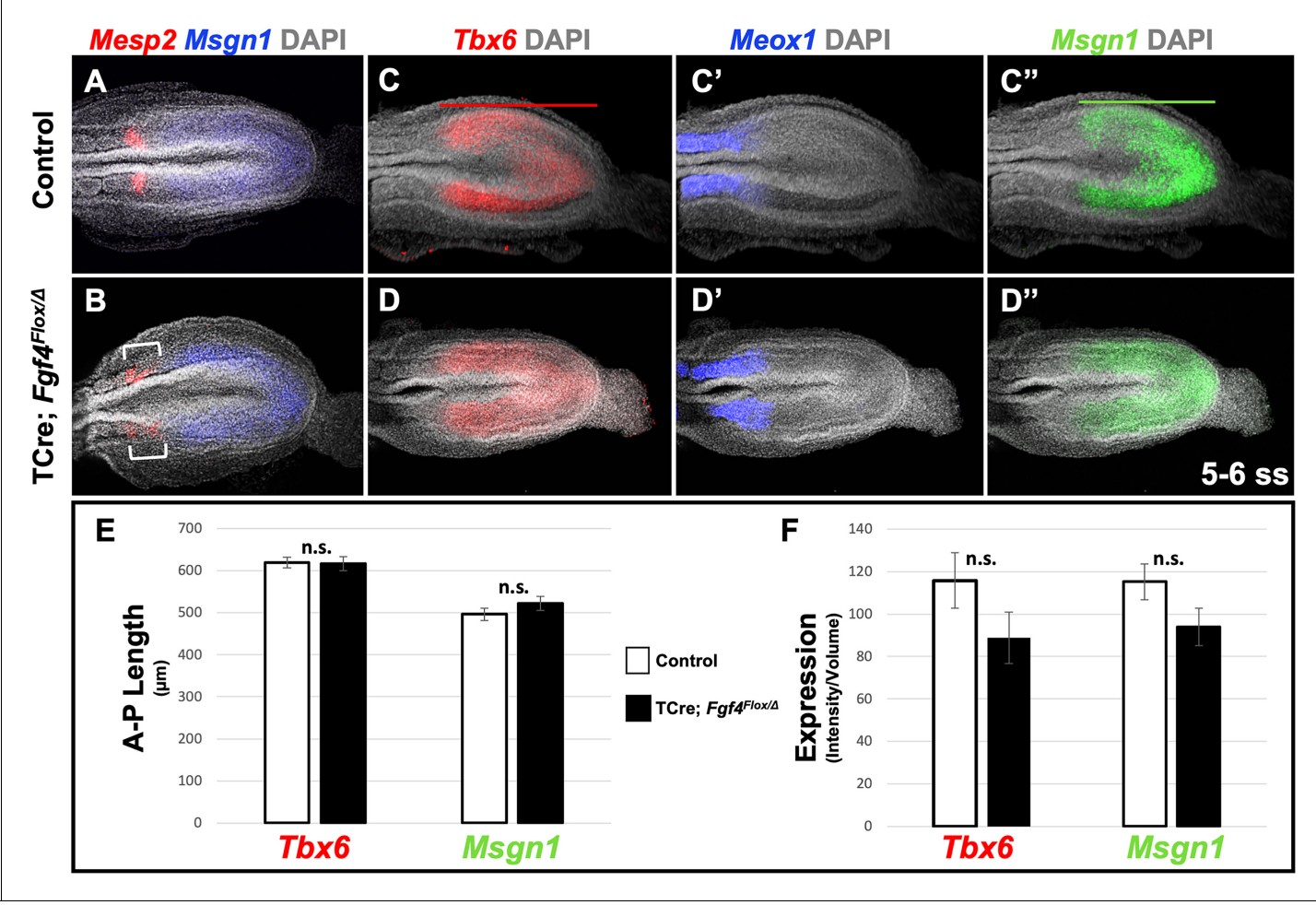

**Figure 2.** *Fgf4* mutants maintain a normal wavefront. (**A, B**) HCR staining of 5–6 somite stage control (A, n = 8) and *Fgf4* mutant embryos (B, n = 11). In mutants, *Msgn1* expression was similar between mutants and controls, but *Mesp2* expression was abnormal (6/11). (**C–D''**) HCR staining of 5–6 somite stage control and *Fgf4* mutant embryos showing normal expression of *Meox1* (control, n = 4; mutant, n = 4), *Msgn1* (control, n = 9; mutant, n = 8), and *Tbx6* (control, n = 10; mutant, n = 8). (**E**) There is no significant difference in the anterior-posterior length of *Tbx6* and *Msgn1* domains in control (n = 9) and mutant embryos (n = 8; green and red bars in C and C'). (**F**) Quantification of *Tbx6* and *Msgn1* mRNA expression, determined by measurement of fluorescence intensity per cubic micrometer within the volume of the *Msgn1* or *Tbx6* domain of expression, respectively. There is no significant difference between control and mutant embryos. In E and F, data are mean ± s.e.m, significance determined by a Student's t-test. *Msgn1*: control n = 7; mutant n = 8. *Tbx6*: control, n = 9; mutant, n = 8. All images: MIP, dorsal view, anterior left. Same embryo shown in C-C'' and D-D''.

## Wavefront gene expression is normal in *Fgf4* mutants

For the rest of our mRNA expression analysis, we mostly relied on whole mount in situ HCR in conjunction with Ce3D+ tissue clearing. Because the fluorescent signal intensity corresponds proportionally to the number of mRNA molecules (*Trivedi et al., 2018*), HCR allows for quantitative and multiplexed mRNA detection (*Choi et al., 2018*). Accurate quantification of the HCR signal requires a method such as Ce3D+ tissue clearing (see Materials and methods), which allows for imaging in deep tissues without signal loss.

We first addressed why a distinct and symmetrical band of *Mesp2* expression fails to form in *Fgf4* mutants. *Mesp2* expression is normally triggered by activated Notch (NICD) and TBX6 at the determination front in the anterior PSM, where cells fall below a critical threshold of wavefront FGF signaling (*Oginuma et al., 2008*). We previously showed that this FGF activity was encoded by *Fgf4* or *Fgf8*; if we inactivated both *Fgfs* with TCre (*Fgf4/Fgf8* double mutant), *Mesp2* expression is transiently activated throughout the PSM, which then aberrantly expresses the paraxial mesoderm marker, *Meox1* (*Naiche et al., 2011*). To determine if the wavefront position is altered in *Fgf4* mutants we examined the anterior-posterior length and expression levels of genes responsive to this

activity at early somite stages when aberrant *Mesp2* expression occurs. Both *Msgn1* and *Tbx6* are required to specify the paraxial mesoderm (*Chapman and Papaioannou, 1998*; *Nowotschin et al., 2012*; *Chalamalasetty et al., 2011*), and are silenced in the absence of wavefront activity in *Fgf4/ Fgf8* double mutants (*Naiche et al., 2011*). Neither the length nor level of expression of *Tbx6* and *Msgn1* were significantly altered in *Fgf4* mutants (*Figure 2C,C"*, *D,D"*, *E,F*). Consistent with this observation, the paraxial differentiation marker, *Meox1*, was not expanded into the PSM (*Figure 2C' and D'*), as occurs in *Fgf4/Fgf8* double mutants (*Naiche et al., 2011*).

Therefore, we conclude that wavefront activity is normal in *Fgf4* mutants, an insight consistent with the observation that normal axis extension occurs in these mutants, with no loss of caudal vertebrae (*Anderson et al., 2016a*). We surmise that this normal wavefront position and signaling in *Fgf4* mutants are likely maintained by *Fgf8*, which is expressed at normal levels (*Figure 3A–C*) and is sufficient for maintaining normal expression levels of the FGF target genes, *Spry2*, *Etv4*, and *Spry4* (*Figure 3D–F'''*); these FGF targets are silenced in *Fgf4/Fgf8* double mutants (*Naiche et al., 2011*). Therefore, neither a change in wavefront activity nor a change in determination front position explains the aberrant pattern of *Mesp2* expression in *Fgf4* mutants.

## Oscillation of Notch family components is altered in the PSM of *Fgf4* mutants

We then examined the pattern of activated Notch in *Fgf4* mutants by immunostaining for Notch intracellular domain (NICD), which is an obligate factor in the transcriptional complex that activates *Mesp2* (*Takahashi et al., 2000*). Overall NICD levels were unchanged between *Fgf4* mutants and littermate controls at the embryonic stages when somitogenesis was abnormal (*Figure 4A*). At these stages, we always observed two to three distinct stripes of NICD along the anterior-posterior axis in

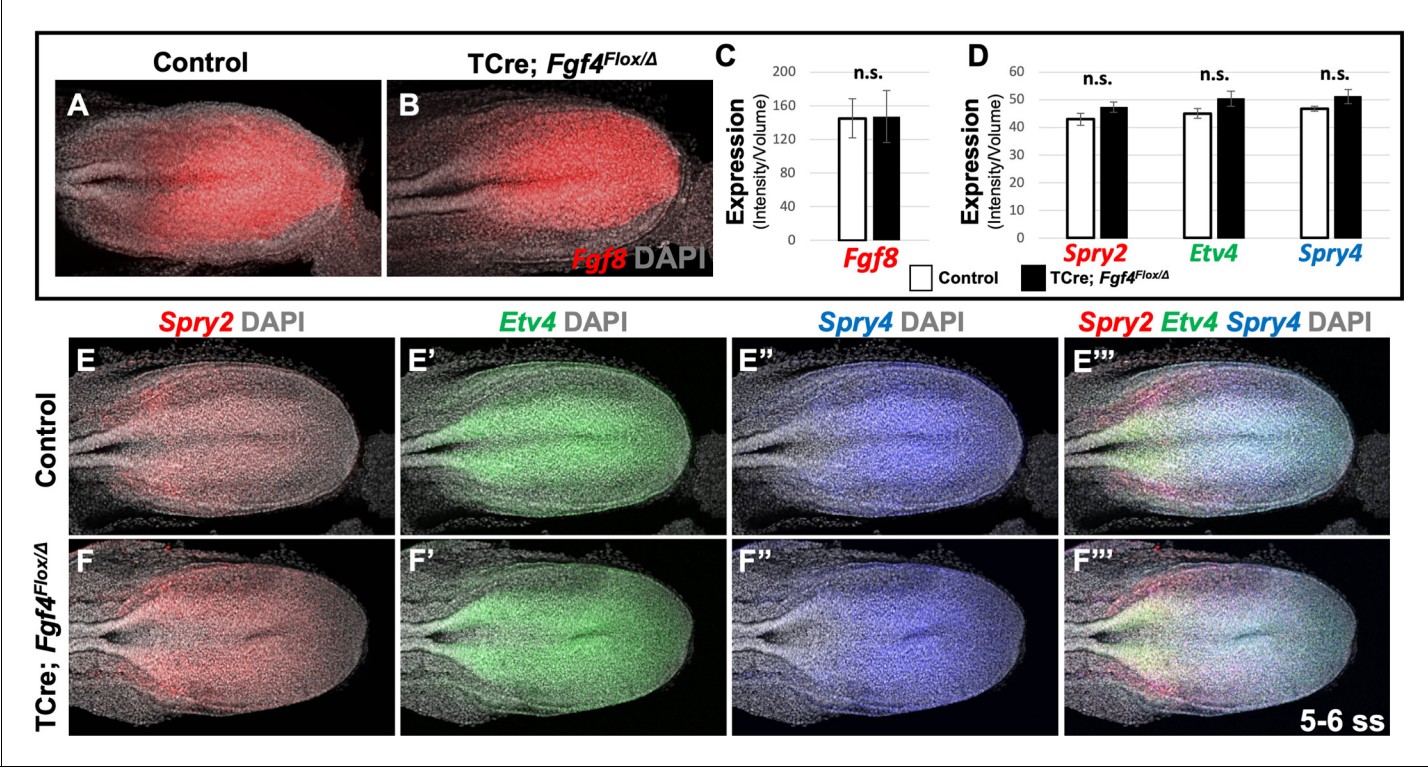

**Figure 3.** *Fgf8* is sufficient for maintaining canonical FGF-responsive gene expression in *Fgf4* mutants. (A, B) HCR staining of 5–6 somite stage control (n = 6) and *Fgf4* mutant (n = 3) embryos showing comparable expression of *Fgf8,* quantified in (C). (C) Quantification of *Fgf8* mRNA expression, determined by measurement of fluorescent intensity per cubic micrometer within the volume the mesoderm- specific *Tbx6* expression domain. Note, there is no significant difference between control and mutant embryos. (D) Quantification of HCR analysis of *Spry2*, *Etv4*, and *Spry4* expression in 5–6 somite stage control (n = 4) and *Fgf4* mutant (n = 4) embryos imaged in (E–F'''). Data in C and D are mean ± s.e.m, significance determined by a Student's t-test. All images: MIP, dorsal view, left. Same embryo shown in E- E''' and F- F'''.

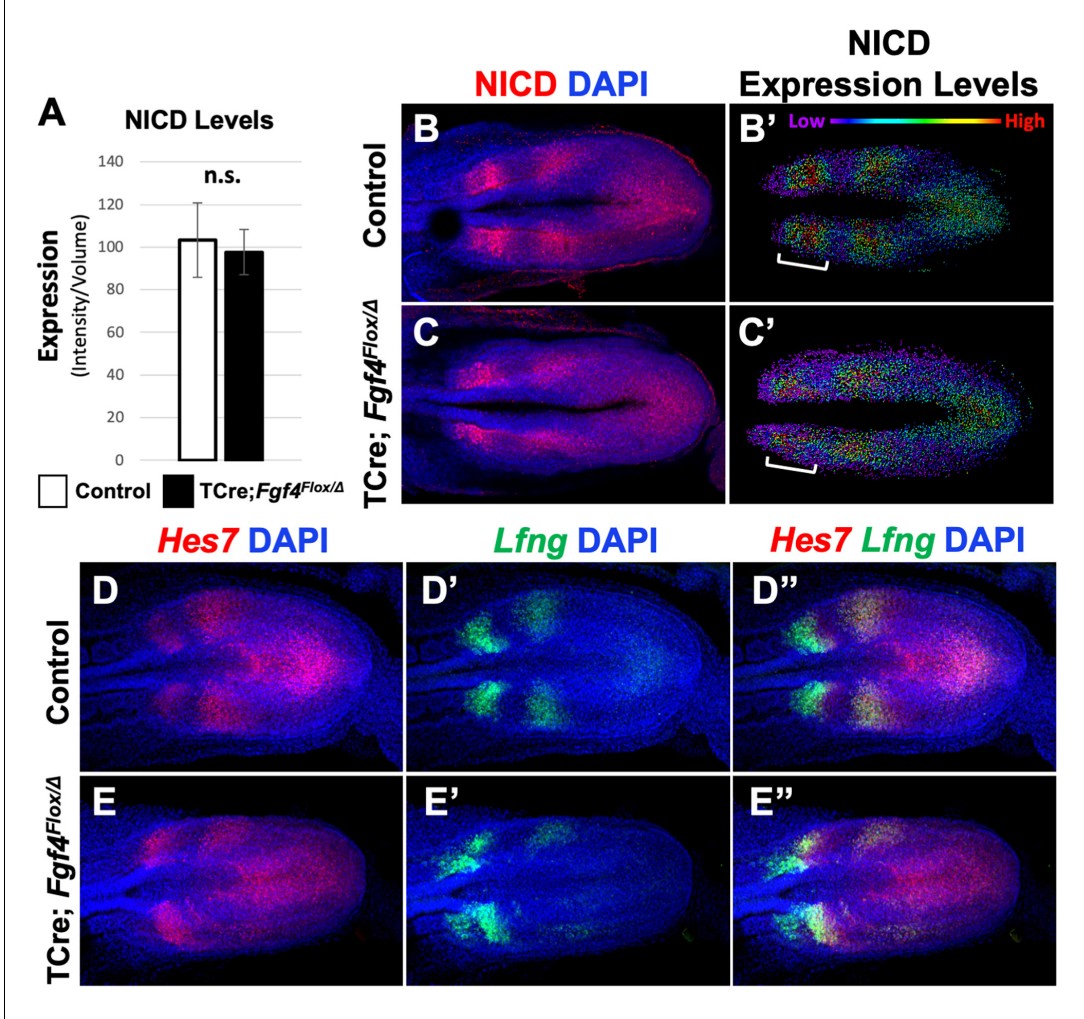

**Figure 4.** Pattern of Notch signaling is abnormal in *Fgf4* mutants. (A) Quantification of immunostained NICD fluorescent signal specifically in the mesoderm of 5–6 somite stage embryos. Note, there is no significant difference between *Fgf4* mutants (n = 5) and littermate controls (n = 5). Data are mean ± s.e.m, significance determined by a Student's t-test. (B, C) 5–6 somite stage control (B) or *Fgf4* mutant (C) embryos immunostained for NICD. (B' C') Same images as in (B,C) but visualized using the Imaris spot modeling function where each fluorescent signal is represented as a colored sphere according to pixel intensity (lowest value is purple and highest value is red, as indicated). Note that the pattern in the *Fgf4* mutant (C') is less distinct compared to littermate control (B'); brackets indicate anterior PSM. (D–E'') HCR staining of representative 5–6 somite stage control (n = 10) and *Fgf4* mutant (n = 8) embryos for the indicated genes. Note *Fgf4* mutant expression of *Hes7* and *Lfng* in mutants is indistinct. All images: MIPs, dorsal view, anterior left. Same embryo shown in B, B'; C, C'; D-D''; E-E''.

The online version of this article includes the following figure supplement(s) for figure 4:

**Figure supplement 1.** *Lfng* expression is abnormally patterned in *Fgf4* mutants.

both mutant and littermate control PSM (*Figure 4B–C*), demonstrating the oscillatory nature of Notch activation (*Bone et al., 2014*; *Morimoto et al., 2005*; *Huppert et al., 2005*). However, mutant oscillatory stripes of NICD were always less ordered, compared to controls. This was more evident when the relative intensities of NICD immunostaining signals were modeled using Imaris software. In controls, cells medial-lateral to each other had similar levels of activated NICD, whereas in mutants, this organization was less distinct (*Figure 4B'–C'*). Importantly, this blurred pattern occurred in the anterior PSM, where NICD activates *Mesp2* expression at the determination front (*Figure 4B'–C'*, brackets).

A synchronous oscillatory NICD pattern is achieved through negative-feedback loops controlled by *Lfng Morimoto et al., 2005*) and *Hes7* (*Niwa et al., 2011*). *Lfng* encodes a glycosyltransferase that represses Notch signaling in the PSM (*Ferjentsik et al., 2009*; *Dale et al., 2003*; *Okubo et al.,*

2012), possibly via glycosylation of DLL1 and/or DLL3 (*Panin et al., 2002*; *Serth et al., 2015*). *Hes7* is a basic helix-loop-helix transcription factor that is essential for repression of a number of Notch target genes (*Niwa et al., 2007*; *Bessho, 2001*; *Chen et al., 2005*). *Hes7* and *Lfng* are themselves targets of activated Notch and therefore their expression oscillates (*Niwa et al., 2011*). Disruption of either gene causes uncoordinated Notch signaling in the PSM (*Niwa et al., 2011*).

Although oscillatory expression of *Hes7* and *Lfng* are apparent in both *Fgf4* mutants and littermate controls (*Figure 4D–E''*), the mutant patterns are indistinct, similar to the mutant NICD pattern (*Figure 4D–E''*). *Lfng* expression is particularly disordered in the anterior PSM (*Figure 4E' and E''*; *Figure 4—figure supplement 1*), where both NICD and *Mesp2* also show aberrant patterning (*Figure 4C,C'* and *Figures 1L* and *2B*). *Hes7* expression is likewise disordered in the mutant PSM at these stages (*Figure 4E,E''*). Together, these data suggest the aberrant *Mesp2* pattern in *Fgf4* mutants can be explained by loss of coordinated Notch signaling at the differentiation front.

## *Hes7* expression is reduced in the PSM of *Fgf4* mutants

We proceeded to focus on the expression of *Hes7* in greater detail in *Fgf4* mutants. The transcriptional repressor encoded by *Hes7* is considered to be a fundamental pacemaker of the segmentation clock that controls somitogenesis (*Kageyama et al., 2012*). *Hes7* is within the *Hes/Her* class of transcriptional repressors that are the only oscillating clock genes conserved among mouse, chicken, and zebrafish (*Krol et al., 2011*). In the mouse, mutations that accelerate HES7 production will accelerate the tempo of the segmentation clock (*Harima et al., 2013*). Moreover, HES7 directly represses *Lfng* transcription (*Chen et al., 2005*) and is required for normal *Lfng* oscillations within the PSM (*Ferjentsik et al., 2009*; *Bessho, 2001*; *Dale et al., 2003*; *Okubo et al., 2012*). Therefore, defective *Hes7* expression could account for the abnormal NICD and *Mesp2* patterning in *Fgf4* mutants, possibly by causing aberrations in *Lfng* expression.

Analysis of *Hes7* mRNA in *Fgf4* mutants at 5–6 somite stages, using conventional WISH, revealed a reduced and aberrant expression pattern in *Fgf4* mutants (*Figure 5—figure supplement 1*). Analysis with HCR was much more informative, allowing unambiguous pattern analysis (*Figure 4*) and quantification (*Figure 5*, *Figure 5—figure supplement 2*, *Figure 5—figure supplement 3*). Characterization of an oscillatory gene expression pattern in the PSM is achieved by classifying static expression patterns into three phases (*Pourquié and Tam, 2001*). This approach has been used to characterize *Hes7* expression at E9.5-E10.5, when one to two stripes of expression are observed (*Bessho, 2001*; *Bessho et al., 2003*). However, in the E8.5 PSM, when aberrant Notch signaling occurs in *Fgf4* mutants, there are two to three *Hes7* expression stripes in both controls and *Fgf4* mutants (*Figures 4* and *5*). Therefore, we generated criteria for classification of phases at E8.5 as follows. Phase I: three stripes with the most posterior stripe limited to the posterior midline and the most anterior stripe having reached the anterior boundary of the PSM (*Figure 5A*). Phase II: two stripes with the posterior stripe having expanded laterally compared to phase I and the anterior stripe having not reached the anterior limit of the PSM (*Figure 5B*). Phase III: two distinct stripes and a third stripe initiating at the posterior midline (*Figure 5C*). Control embryos are distributed nearly equally between each phase (*Figure 5G*) as is the case for *Hes7* expression in older embryos (*Bessho, 2001*; *Bessho et al., 2003*). It is challenging to place *Fgf4* mutants in phases, emphasizing the aberrant *Hes7* expression pattern. However, in our assessment, we allocated *Fgf4* mutants within all three phases (*Figure 5D–F*), with most found in oscillation phase II (*Figure 5G*).

*Hes7* is expressed in both the PSM and neural ectoderm. To quantify expression only in the PSM, we used Imaris software to create a volumetric model of the PSM, based on *Tbx6* expression, which is limited to the PSM at these stages (*Chapman et al., 1996*). By measuring the intensity of the *Hes7* HCR signal within this volume, we obtained a PSM-specific *Hes7* quantification for each embryo (see *Figure 5—figure supplement 3* and Material and methods). PSM-specific *Hes7* expression is reduced in *Fgf4* mutants to 51.8% of that expressed in litter-mate controls (*Figure 5H*). A comparison between mutant and control *Hes7* expression levels within each phase reveals that the reduction in *Fgf4* mutants is not due to the difference in phase allocation between *Fgf4* mutants and littermate controls (*Figure 5—figure supplement 4*). Moreover, after the 22 somite stage, when forming somites are normally segmented in mutants (*Figure 1J*), there is no significant difference in *Hes7* PSM expression levels between *Fgf4* mutants and controls (*Figure 5—figure supplement 5*). Therefore, we conclude that a reduction of *Hes7* expression may account for the defective segmentation during rostral somitogenesis in *Fgf4* mutants.

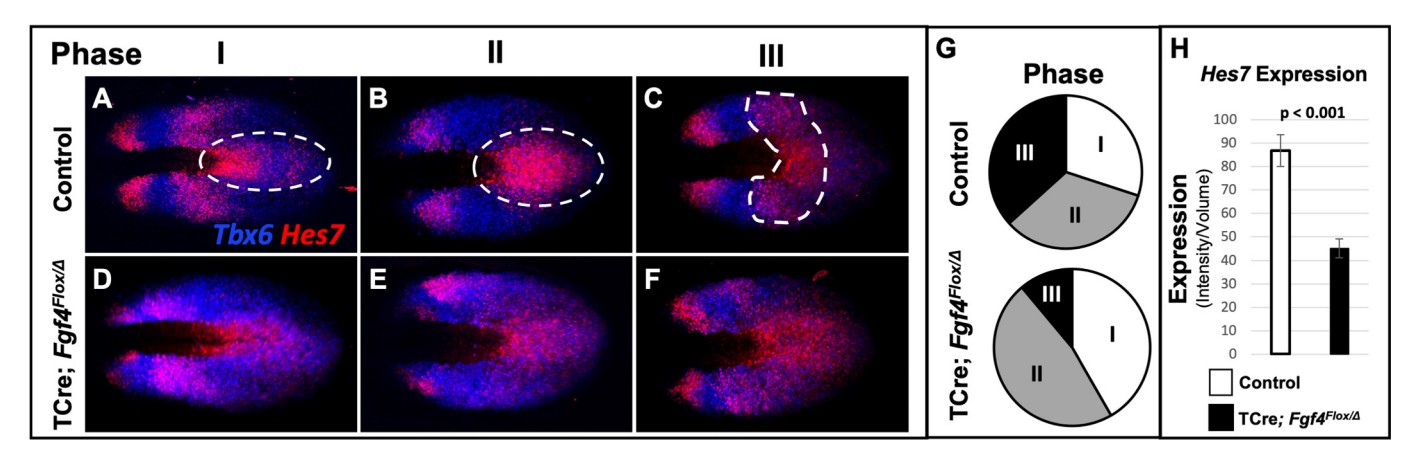

**Figure 5.** *Hes7* expression is abnormally patterned and reduced in *Fgf4* mutants. (**A–F**) HCR staining for *Hes7* expression in 5–6 somite stage control and *Fgf4* mutant embryos sorted by phase of oscillation, showing abnormal expression of *Hes7* in mutants. (**G**) Distribution of phases in control and *Fgf4* mutants shown in A-F; controls: phase I, n = 9; phase II, n = 10; phase III, n = 11. *Fgf4* mutants: phase I, n = 6; phase II, n = 9; phase III, n = 2. (**H**) Quantification of *Hes7* expression specifically within the PSM (all phases combined) in 5–6 somite stage control (n = 11) and *Fgf4* mutant (n = 8) embryos shows significantly decreased expression in the *Fgf4* mutant. Quantification of *Hes7* mRNA expression, determined by measurement of fluorescence intensity per cubic micrometer within the volume the mesoderm-specific *Tbx6* expression domain. Data in H are mean ± s.e.m, significance determined by a Student's t-test. All images are MIPs, dorsal view, anterior to left.

The online version of this article includes the following figure supplement(s) for figure 5:

**Figure supplement 1.** *Hes7* expression is abnormal in *Fgf4* mutants.

**Figure supplement 2.** Using *Hes7* null homozygotes to adjust Imaris baseline cutoff.

**Figure supplement 3.** Imaris-modeling HCR Expression.

**Figure supplement 4.** *Hes7* expression in *Fgf4* mutants is reduced, irrespective of oscillation phase.

**Figure supplement 5.** *Hes7* expression is normal at 24–26 somite stage in *Fgf4* mutants.

To correlate *Hes7* expression levels with pattern, we created Imaris-generated spot models of the HCR signal, which correspond to a single mRNA molecule or clusters of mRNA molecules within a subcellular-volume (see Materials and methods). We then colored-coded individual spots to reflect the intensity of the *Hes7* HCR signal, using a linear series of intensity-based cutoffs every 20% to generate five colors (quintiles) (*Figure 6A–B*). In controls, the lowest intensity quintile (blue) contains the most spots (40%) and each higher-intensity group contains 10% fewer spots (*Figure 6A*, white bars in C). The distribution of spots in the *Fgf4* mutant, compared to controls, is significantly skewed towards the lowest quintile at the cost of higher-level expression spots (*Figure 6B*, black bars in C). This modeling provides an effective illustration of the pattern of *Hes7* expression.

In control embryos, each stripe of *Hes7* expression contains a concentric gradient of signal intensity with highest expression at the center (white spots) and lowest expression at the outside (blue spots) (*Figure 6A,D*). *Fgf4* mutants maintain some of this concentric organization, but with a greater variability of expression between neighboring cells, the boundaries between groups is less clear (*Figure 6B,E*). In particular the trough between peaks of *Hes7* expression is more shallow, and contains more higher-intensity spots (mostly second quintile, green) than littermate controls (*Figure 6D' and E'*). To determine if this indicated a failure to repress *Hes7* transcription in these regions, we performed HCR using a probe that hybridizes to the *Hes7* intronic sequences, and therefore specific to newly transcribed pre-mRNA. This analysis revealed that transcription of *Hes7* is reduced in the posterior PSM and is more widespread, extending into the trough between *Hes7* mRNA peaks in the *Fgf4* mutant (white brackets in *Figure 6F', G'*). Therefore, we hypothesized that FGF4 is required to maintain *Hes7* transcription in the PSM above a threshold required for normal patterns of Notch gene oscillation. The reduced level of HES7 in *Fgf4* mutants is insufficient to fully autorepress its own expression, resulting in an aberrant pattern of expression.

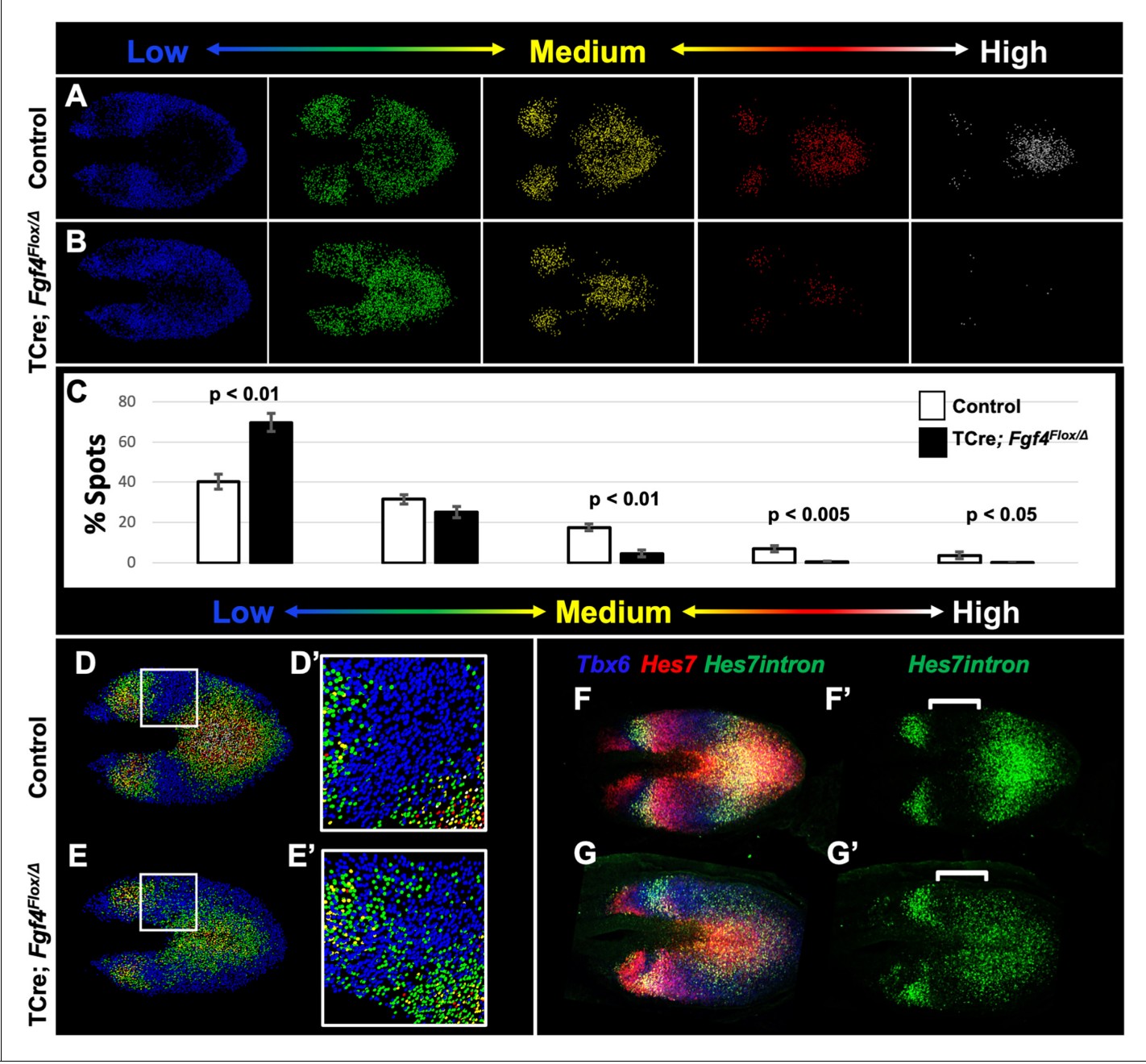

**Figure 6.** Reduced *Hes7* expression with less distinct peaks and troughs in the *Fgf4* mutant PSM. (A, B) Spot models based on HCR analysis of *Hes7* expression, in 5–6 somite stage control and *Fgf4* mutant embryos within the PSM, as defined by the *Tbx6* expression domain. Localized *Hes7* expression is colored by level of expression from low (blue) to high expression (white), as indicated. Note there are less high expressors (yellow, red and white) and more low expressors (blue). This insight is quantified in (C), where the percentage of spots found in each expression-level group is graphed. Data are mean ± s.e.m, significance determined by a Student's t-test; control, n = 6; mutant, n = 5. (D–E') Composite of spot models from A and B. Note that the *Hes7* expression trough (boxed region, expanded in D' and E') between anterior and posterior oscillatory peaks is less distinct in mutant (E') with more higher expressors (green) than in control embryos (D'). (F–G') MIP of HCR staining of 5–6 somite stage control and *Fgf4* mutant embryos using probes against *Tbx6*, *Hes7*, and *Hes7intron* which specifically labels active *Hes7* transcription. Note the trough of *Hes7intron* signal between peaks (brackets) in mutants is less pronounced than controls, indicating a failure of transcriptional repression of *Hes7* in this region. Representative embryos are shown; control, n = 6; mutant, n = 5. All images are dorsal views, anterior left. Same embryo shown in F, F'; G, G'.

## A synergistic defect in *Fgf4/Hes7* mutants reveals that *Fgf4* is required to maintain *Hes7* above a critical threshold

To test our hypothesis that a reduction of *Hes7* expression causes the *Fgf4* mutant vertebral defects, we asked if these defects worsen if we further reduce *Hes7* expression by removing one gene copy. We compared such mutant to littermate controls that were simple *Fgf4* mutants (with two wildtype *Hes7* alleles) or *Hes7* heterozygotes. These *Hes7* heterozygotes also carried TCre and one floxed *Fgf4* allele (see *Table 1*, and *Figure 7*) resulting in *Fgf4* heterozygosity in the TCre expression domain. However, *Fgf4* heterozygosity had no effect on the phenotype because the defects we observed were similar to the *Hes7* heterozygous defects reported by the Dunwoodie lab (*Sparrow et al., 2012*). About 50% of our *Hes7* heterozygotes had defective lower thoracic vertebrae with a frequency of 2.8 defects per animal. In littermate *Fgf4* mutants, vertebral defects were completely penetrant, with an average of 7.7 defects per animal, a frequency similar to that of progeny in our original genetic cross (*Figure 1*). However, compound *Fgf4/Hes7* mutants displayed a large set of defects with 25 defects per animal (*Figure 7C–E*), a frequency significantly greater than littermate *Fgf4* mutants (3-fold greater, p<0.005) or *Hes7* heterozygotes (9-fold greater, p<0.001). These data indicate a synergistic, as opposed to additive, effect of loss of one *Hes7* allele in the *Fgf4/Hes7* compound mutant.

We performed a similar genetic analysis where we determined if *Hes7* heterozygosity likewise affects *Fgf8* mutant defects. We examined littermates of the last cross described in *Table 1* (*Figure 7—figure supplement 1*). *Fgf8* mutants in this cohort displayed the same homeotic transformations we previously observed with incomplete penetrance and expressivity: small ribs in the posterior cervical vertebra (3/5, 2 bilateral and 1 unilateral) or anterior lumbar vertebra (3/5, 1 bilateral and 2 unilateral). *Hes7* heterozygotes (also heterozygous for a floxed *Fgf8* allele) displayed a nearly identical frequency (2.7 defects per animal) as observed in the *Fgf4/Hes7* experiment (*Figure 7A*). Vertebral defects in compound *Fgf8/Hes7* mutants were not synergistic (four defects per animal) but were clearly the *Hes7* heterozygous defects added to the relatively mild *Fgf8* defects (1.8 defects per animal, *Figure 7—figure supplement 1*). Therefore, we demonstrate that the relationship between *Hes7* and *Fgf4* is unique in that there is no genetic interaction between *Fgf8* and *Hes7*.

We then examined levels and spatial patterns of *Hes7* mRNA expression in each genotype from the *Fgf4/Hes7* mutant littermates, specifically within a volumetric model of the *Tbx6* expression domain, as we had in *Fgf4* mutants (*Figures 5* and *6*). We observed a reduction in *Hes7* expression that correlated with the severity of vertebral defects within each genotype (*Figure 7E*). *Hes7* null heterozygotes had only a 19% reduction in expression, presumably because a loss of HES7 auto-repression results in enhanced transcription from the remaining wildtype allele. Such compensatory upregulation fails in compound *Fgf4/Hes7* mutants because we observed an 80% *Hes7* reduction to occur in these embryos (*Figure 7E*). Analysis of intensity-colored spot models of PSMs of these genotypes suggests a threshold effect of *Hes7* levels on oscillatory gene expression (*Figure 7F–I*). Oscillations appear relatively normal if *Hes7* levels are reduced 19% (in *Hes7* heterozygotes, *Figure 7G*) but are disordered with a 33% reduction (in *Fgf4* mutants, *Figure 7H*). The synergistic 80% reduction of *Hes7* that occurs in compound *Fgf4/Hes7* mutants causes a severe dampening of oscillatory gene expression (*Figure 7I*). The resulting vertebral defects, though relatively severe (*Figure 7A–D*), are not as extreme as occurs in *Hes7* null homozygotes (*Bessho, 2001*), indicating that this reduced level of *Hes7* supports limited patterning during segmentation. Together, our data support our hypothesis that FGF4 acts to maintain *Hes7* above a necessary threshold for normal somitogenesis. In *Fgf4* mutants, a reduction in *Hes7* levels, leads to abnormal patterns of activated Notch. Abnormal Notch oscillations initiate *Mesp2* unevenly in the anterior PSM leading to improperly shaped somites and subsequently, malformed vertebrae.

## Discussion

We describe mutants with only *Fgf4* or *Fgf8* inactivated specifically in the primitive streak and PSM. In previous work, we reported that somites form in such mutants (*Naiche et al., 2011*; *Perantoni et al., 2005*); here we examine the patterning of these somites, the Notch pathway oscillations that control their formation, as well as the vertebral elements that they eventually form. We found that *Fgf4* mutants display a range of cervical and thoracic vertebral defects that are caused by

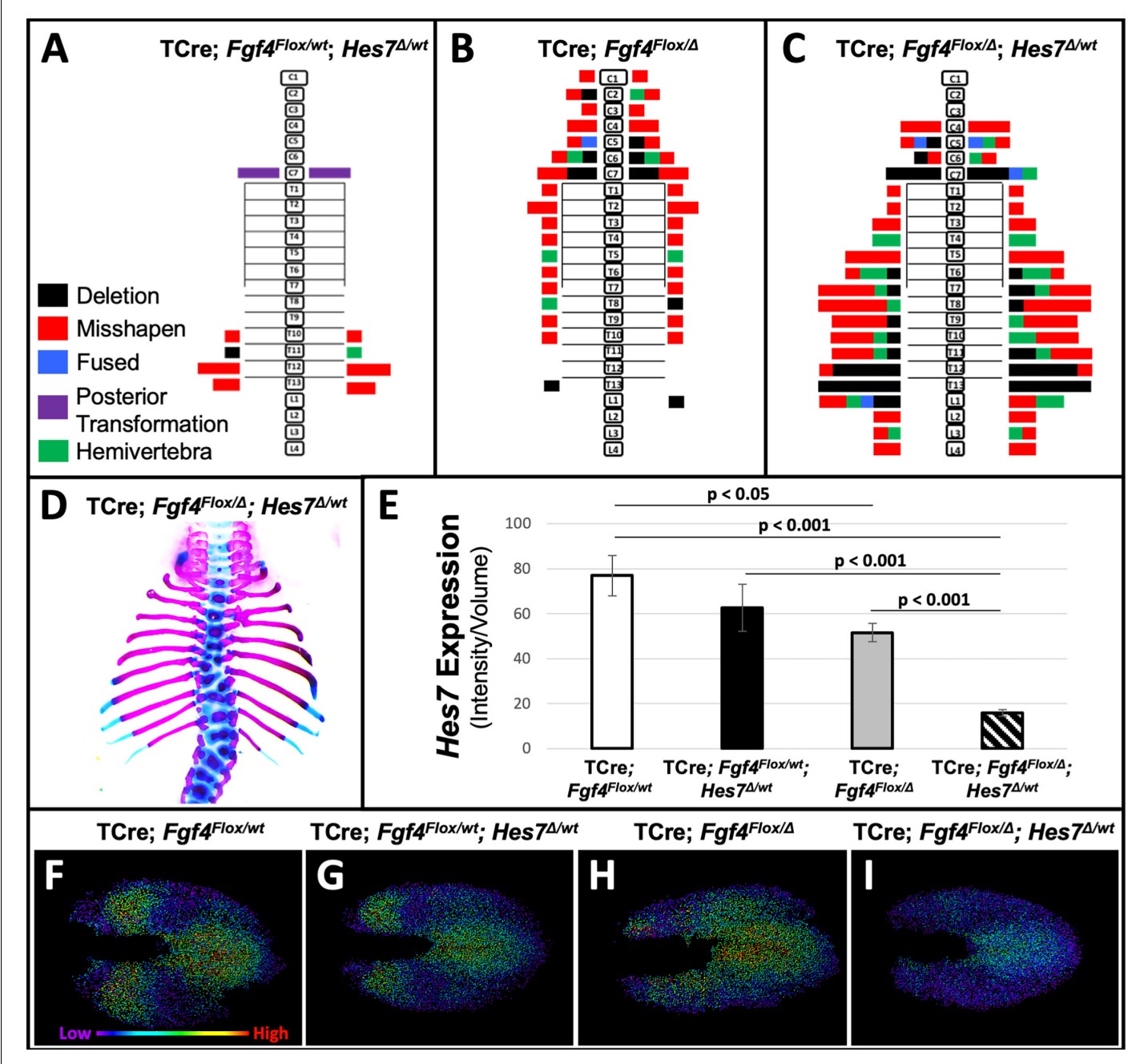

**Figure 7.** Removal of one *Hes7* allele exacerbates vertebral defects in *Fgf4* mutants. (**A–C**) Histograms representing the vertebral column and associated ribs (C = cervical, T = thoracic, L = lumbar), showing variety and location of vertebral defects at E18.5 in the genotypes indicated (A and B, n = 7; C, n = 6). Each block in the histogram represents a single defect as indicated in the key in A. (**D**) Skeletal preparation of a representative E18.5 compound *Fgf4* mutant- *Hes7* heterozygote (ventral view). (**E**) Quantification of *Hes7* expression within the PSM in 5–6 somite stage embryos of the indicated genotype: control (n = 10), *Fgf4-Hes7* double heterozygotes (n = 6), *Fgf4* mutants (n = 10), and compound *Fgf4* mutant- *Hes7* heterozygotes (n = 7). Quantification of *Hes7* mRNA expression, determined by measurement of fluorescent intensity per cubic micrometer within the volume of the mesoderm-specific *Tbx6* expression domain. Data in E are mean ± s.e.m, significance determined by a Student's t-test. (**F–I**) Spot models based on HCR analysis of *Hes7* expression, in embryos of the indicated genotype, within the PSM, as defined by the *Tbx6* expression domain. Localized *Hes7* expression is colored by level of expression from low (purple) to high expression (red), as indicated. Images F-I are dorsal views, anterior left. The online version of this article includes the following figure supplement(s) for figure 7:

**Figure supplement 1.** Removal of one *Hes7* allele has no effect on vertebral defects in *Fgf8* mutants.

aberrant Notch oscillations during somitogenesis. In contrast, the vertebral columns of *Fgf8* mutants are normally segmented, with about 30% displaying minor homeotic transformations; such alternations in vertebral identity are not likely due to defects in somitogenesis per se. Previously, a role for FGF signaling in wavefront activity in chick and zebrafish embryos was supported by pharmacological manipulation (*Dubrulle et al., 2001*; *Sawada et al., 2001*) and in mouse mutants where both *Fgf4* and *Fgf8* are simultaneously inactivated using the same TCre activity that we use in this study (*Naiche et al., 2011*). These double *Fgf4/Fgf8* mutants as well as mutants with tissue-specific inactivation of *Fgfr1* support a role for FGF signaling in clock oscillations, but these conclusions were complicated by a loss of PSM tissue due to potential wavefront defects (*Oginuma et al., 2008*; *Naiche et al., 2011*; *Wahl et al., 2007*; *Niwa et al., 2007*). Compared to these *Fgf* pathway mutants, the phenotype of *Fgf4* mutants is relatively subtle, with no wavefront defect, as indicated by no loss of caudal vertebrae (*Anderson et al., 2016a*) and no quantitative change in wavefront gene markers (*Figure 2*). This lack of any overt axis extension defect allows us to unambiguously identify *Fgf4* as an *Fgf* ligand gene required for a normal segmentation clock. This insight is supported by recently published work showing that FGF signaling regulates the dynamics of gene expression oscillations in PSM cells cultured in vitro (*Diaz-Cuadros et al., 2020*).

Our mutants model human Segmentation Defects of the Vertebrae (SDV); *Fgf4* mutants resemble CS and the synergistic phenotype resulting from the additional removal of one *Hes7* copy in these mutants resembles human SCDO. Both of these diseases are caused by mutations in Notch signaling components that we find are misexpressed in *Fgf4* mutants, such as *HES7*, *MESP2*, and *LFNG* (*Eckalbar et al., 2012*; *Giampietro et al., 2009*). With regard to the human FGF pathway and SDV, a straightforward gene-disorder relationship has not been uncovered, probably because of the pleiotropic phenotypes of such putative mutations would preclude embryonic survival. Sparrow et al found that gestational hypoxia in mice results in an increase in the severity and penetrance of CS in *Notch1*, *Mesp2*, and *Hes7* heterozygotes (*Sparrow et al., 2012*). Intriguingly, CS is more severe in children living at high altitudes (*Hou et al., 2018*), suggesting a similar environmental effect may affect human development. In mice, hypoxia was found to diminish FGF signaling components, but expression of *Fgf8* was unchanged, and *Fgf4* was unexamined (*Sparrow et al., 2012*). Our data suggest that FGF4 may be part of the system that is responsive to this environmental insult. However, if this is the case, reduced FGF4 activity cannot be the only response to reduced oxygen levels because gestational hypoxia reduces expression of the FGF target gene *Spry4* in the PSM, whereas we found no change in expression in this or other such canonical FGF target genes in *Fgf4* mutants.

Rather, as is the case for hypoxia-treated embryos (*Sparrow et al., 2012*), *Hes7* expression is reduced in *Fgf4* mutants and we propose that this leads to aberrant segmentation. An indispensable tool to this insight was the use of multiplex HCR imaging (*Choi et al., 2014*; *Choi et al., 2018*; *Trivedi et al., 2018*). The simultaneous fluorescent imaging of multiple mRNA domains with this technique is particularly useful in a complex embryonic process, such as somitogenesis, where many complex and dynamic gene expression patterns need to be analyzed. HCR is a relatively new tool for mutant embryo analysis, just beginning to be used to analyze mutant embryos (*Lignell et al., 2017*; *Kim et al., 2019*; *Attardi et al., 2018*; *McKinney et al., 2016*). We used HCR to examine the expression of multiple genes in a single tissue by combining it with Imaris image analysis software. This combination of HCR and Imaris-modeling is broadly applicable for in situ quantification of gene expression within any complex embryonic or clinical sample. With distinct molecular markers, one can use this approach to quantify gene expression in any cell population with a precision that heretofore was not possible and still retain the intact whole mount embryo or tissue. Here, we generated and analyzed Imaris-based volumetric models of the PSM, based on *Tbx6* expression, and determined that *Hes7* levels are reduced by about 40–50% specifically in the PSM of *Fgf4* mutants during early somite stages when the progenitors of aberrant vertebrae are segmenting.

We propose that this reduction in *Hes7* levels is the primary defect in *Fgf4* mutants, and subsequently causes an irregular activated Notch (NICD) pattern, misexpression of *Mesp2*, and ultimately leads to vertebral defects. A reduction in *Hes7* levels, due to heterozygosity or hypomorphic mutations, has been shown to cause vertebral defects (*Sparrow et al., 2012*; *Fujimuro et al., 2015*; *Stauber et al., 2009*). The concept that *Hes7* is generally regulated by FGF signaling is supported by numerous studies (*Naiche et al., 2011*; *Niwa et al., 2007*; *Harima and Kageyama, 2013*). Future work will determine whether *Hes7* is a direct FGF4 signaling target, possibly through putative ETS binding sites in the *Hes7* promoter (*González et al., 2013*). Alternatively, FGF4 may indirectly

regulate *Hes7* transcription by affecting the activity of TBX6, NICD or MESOGENIN, all shown to regulate *Hes7* expression in the PSM (*González et al., 2013*; *Hayashi et al., 2018*). If so, we speculate such regulation would be post-transcriptional because we find no significant difference in levels of the genes encoding these factors. Additionally, it is possible that FGF4 signaling is interfacing with WNT signaling, which also has been shown to regulate *Hes7 Aulehla et al., 2008*; *Dunty et al., 2008*; *González et al., 2013*. Given the molecular feedbacks during genetic oscillations in the PSM, we considered whether FGF4 might have a primary role on *Lfng* expression, whose pattern is abnormal in our mutants. However, we suggest this is not the case, because, although embryos completely lacking *Lfng* also display aberrant *Hes7* oscillations (*Niwa et al., 2007*; *Stauber et al., 2009*), overall *Hes7* mRNA levels are not reduced (*Ferjentsik et al., 2009*), unlike the case in our *Fgf4* mutants.

We support our proposal that the observed 50% reduction in *Hes7* is the cause of the *Fgf4* mutant with the synergistic worsening of the vertebral segmentation defect that occurs when we further reduce *Hes7* levels by removal of one wildtype allele. Notably, such synergy in defects does not occur when a copy of *Hes7* is removed from *Fgf8* mutants (*Figure 7—figure supplement 1*). From another perspective, the mild CS caused by loss of one *Hes7* allele in a wildtype background (*Sparrow et al., 2012*) is worsened 9-fold by removing *Fgf4* (*Figure 7*). Therefore, *Fgf4* is a robustness-conferring gene that buffers somitogenesis against a perturbation in *Hes7* gene dosage (*Félix and Barkoulas, 2015*).

We observed that in *Fgf4* mutants *Hes7* mRNA levels are reduced during rostral somitogenesis at E8.5 (5–6 somite stages), when defective somitogenesis occurs. At about E9.5 (24–26 somite stages), *Hes7* levels are unaffected, and the *Fgf4* mutant embryo is beginning to generate correctly patterned somites that will differentiate into normally patterned vertebrae. *Tam, 1981* showed that between these two stages of mouse development, somites more than double in size. Such an increase in segment size can be due to a slower segmentation clock and/or faster regression of the determination front (*Oates et al., 2012*). *Gomez et al., 2008* found that the caudal movement of the wavefront does not significantly change at these stages, suggesting a slowing clock between E8.5 and E9.5. Consistent with a slower clock, we observed at least 2 and frequently 3 bands of *Hes7* expression at the *Fgf4*-sensitive stages (5–6 somite stages; *Figures 4* and *5*), but only 1 to 2 bands when *Fgf4* loss has no effect on *Hes7* expression (24–26 somite stages; *Figure 5—figure supplement 5*). In future experiments, we will test this hypothesis directly by measuring the clock rate, using recently generated fluorescent transgenic reporters of *Hes7* (*Yoshioka-Kobayashi et al., 2020*). Thus, it appears that FGF4 activity is required to maintain *Hes7* mRNA levels above a certain threshold when the segmentation clock is faster; when Notch oscillations slow, they may become FGF-independent, or other FGFs may be at play. If this is the case in all vertebrates, we might expect embryos with faster segmentation clocks, such as snakes (*Gomez et al., 2008*), to have a longer window of FGF4-dependence for normal somitogenesis.

This study and past work indicate that FGF4 and FGF8 have both shared and unique roles in the vertebrate embryo's posterior growth zone. They redundantly encode wavefront activities, preventing PSM differentiation (*Naiche et al., 2011*; *Boulet and Capecchi, 2012*). In this capacity, when signaling together, they activate downstream canonical FGF target genes, such as *Spry2*, *Spry4*, and *Etv4* (*Naiche et al., 2011*). Here, we show the role that FGF4 signals play in maintaining *Hes7* mRNA levels cannot be replaced by FGF8. Future efforts will explore how the signaling machinery within the PSM discriminates between these two FGF ligands. For example, our work argues for a reassessment of FGFR function in the PSM; prior work, in which the same Cre driver used here (TCre) inactivates *Fgfr1* (*Wahl et al., 2007*), thought to be the only FGF receptor gene expressed in the PSM, does not exactly phenocopy the defects due to TCre-mediated loss of either *Fgf4* (this study) or *Fgf4/Fgf8* (*Naiche et al., 2011*).

Our insights regarding the roles of FGF4 and FGF8 in the caudal embryonic axis are intriguing in the context of the evolution of the vertebrate body plan. Vertebrates, together with Cephalochordates and Urochordates make up the phylum Chordata (*Putnam et al., 2008*). Somitogenesis predates vertebrate evolution as it occurs in Cephalochordata, the most basal living chordates (*Holland and Onai, 2012*). In both Cephalochordates and Urochordates, the FGF essential for embryonic axis extension is an *Fgf8* ortholog (*Bertrand et al., 2015*; *Pasini et al., 2012*), suggesting that this *Fgf* ('*Fgf8/17/18*') may be the ancestral gene in this process. However, *Fgf8/17/18* activity in these invertebrate chordates does not control gene oscillations as these embryos apparently lack

a segmentation clock (*Bertrand et al., 2015*; *Onai et al., 2015*). Therefore, we speculate that the recruitment of an *Fgf4* role during the evolution of vertebrate axis extension may have coincided with the development of gene oscillations that control somitogenesis.

# Materials and methods

## Key resources table

| Reagent type (species) or resource | Designation | Source or reference | Identifiers | Additional information |
|---|---|---|---|---|
| Strain, strain background *Mus musculus* | Fgf4Flox/delta | PMID:10802662 | MGI:95518 | |
| Strain, strain background *Mus musculus* | Fgf8flox/delta | PMID:9462741 | MGI:99604 | |
| Strain, strain background *Mus musculus* | Hes7delta | PMID:11641270 | MGI:2135679 | |
| Strain, strain background *Mus musculus* | TCre | PMID:16049111 | MGI:3605847 | |
| Chemical compound, drug | Alcian Blue 8GX | Sigma | A5268-25G | |
| Chemical compound, drug | Alizarin Red | Sigma | A-5533 | |
| Chemical compound, drug | BM Purple | Roche | 11 442 074 001 | |
| Chemical compound, drug | Dapi | Life Technologies | D21490 | |
| Chemical compound, drug | Iohexol (Nycodenz) | Accurate Chemical and Scientific Corp | AN1002424 | |
| Chemical compound, drug | Ultra-low gelling temperature agarose | Sigma | A5030 | |
| Chemical compound, drug | Hydrogen peroxide | Sigma | 516813–500 ML | |
| Chemical compound, drug | Ammonium Chloride | Mallinckrodt | 3384 | |
| Chemical compound, drug | N-Methylacetamide | Sigma | AN1002424 | |
| Chemical compound, drug | Triton-X-100 | Sigma | T8787 | |
| Chemical compound, drug | fluorescein | Sigma | 46960 | |
| Chemical compound, drug | acid blue 9 | TCI | CI42090 | |
| Chemical compound, drug | rose bengal | Sigma | 198250 | |
| Commercial assay or kit | V3.0 HCR Kit | Molecular Instruments | Wholemount Mouse Embryo | |
| Software, algorithm | Imaris Software | Bitplane Inc | Imaris V9.2.1 | |
| Antibody | Anti-NICD (rabbit polyclonal) | CST | #4147 | (1:100) |
| Antibody | Anti-rabbit alexa-647 (goat polyclonal) | Thermo Fisher Scientific | A32733 | (1:100) |

*Continued on next page*

*Continued*

| Reagent type (species) or resource | Designation | Source or reference | Identifiers | Additional information |
|---|---|---|---|---|
| Antibody | Anti-Dig (sheep FAB fragments) | Roche | 11 093 274 910 | (1:4000) |
| Other | Fetal calf serum | Thermo Fisher Scientific | 16140071 | |

## Alleles, Breeding and Genotyping

All mice were kept on a mixed background. All genetic crosses are shown in *Table 1*, with the female genotype shown first and the male genotype shown second. PCR-genotyping for each allele was performed using the following primer combinations, *Fgf4flox* (*Sun et al., 2000*) (5'- CAGAC TGAGGCTGGACTTGAGG and 5'- CCTCTTGGGATCTCGATGCTGG), *Fgf4Δ* (*Sun et al., 2000*) (5'-CTCAGGAACTCTGAGGTAGATGGGG and 5'-ATCGGATTCCACCTGCAGGTGC), *Fgf8flox* (*Meyers et al., 1998*) (5'-GGTCTTTCTTAGGGCTATCCAAC and 5'-GCTCACCTTGGCAATTAGC TTC) *Fgf8Δ* (*Meyers et al., 1998*) (5'-CCAGAGGTGGAGTCTCAGGTCC and 5'-GCACAAC TAGAAGGCAGCTCCC), *Hes7Δ* (*Bessho, 2001*) (5'-AGAAAGGGCAGGGAGAAGTGGGCGAGC-CAC, and 5'-TTGGCTGCAGCCCGGGGGATCCACTAGTTC), TCre (*Perantoni et al., 2005*) (5'-GGGACCCATTTTTCTCTTCC, and 5'-CCATGAGTGAACGAACCTGG).

## HCR

HCR fluorescent in situ analysis where carried out as described (*Choi et al., 2018*) with the modification of using 60 pmol of each hairpin per 0.5 mL of amplification buffer. Hairpins where left 12–14 hr at room temperature for saturation of amplification to achieve highest levels of signal to noise (*Trivedi et al., 2018*). Stained embryos were soaked in DAPI solution (0.5 μg/mL DAPI in 5x SSC with 0.1% TritonX-100, 1% Tween20) overnight at room temperature. Split initiator probes (V3.0) were designed by Molecular Instruments, Inc.

## Whole mount immunohistochemistry, colorimetric WISH, and skeletal staining

Notch Intracellular Domain staining (NICD) was performed following a significantly modified protocol (*Geffers et al., 2007*); briefly, embryos were dissected in cold PBS, briefly fixed for 5 min in 4% PFA, then fixed for 35 min in equal parts DMSO:Methanol:30% Hydrogen Peroxide at room temperature. They were then rinsed 3 × 5 min in 50 mM Ammonium Chloride at room temperature, blocked in TS-PBS (PBS, 1% Triton-X-100, 10% Fetal Calf Serum) for 30 min at room temperature, then incubated overnight in 1:100 anti-NICD (CST #4147) in TS-PBS at 4° C rocking. The next day, embryos were washed 4 × 10 min in TS-PBS, then incubated overnight in anti-rabbit-alexa647 (ThermoFisher A32733) 1:100 in TS-PBS at 4°C rocking. Embryos were then washed 4 × 10 min in TS-PBS then soaked overnight in DAPI, as described above in the HCR section, and embedded and cleared as described below. WISH and skeletal staining were performed as previously described (*Anderson et al., 2016b*).

## Embedding and clearing

### Embedding

Stained embryos where mounted in coverslip bottomed dishes suspended in ultra-low gelling temperature agarose (Sigma, A5030) that had been cooled to room temperature. Once correct positioning of embryos was achieved the dishes were moved to ice to complete gelling.

### Clearing

For tissue clearing we utilized Ce3D+, a modified version of Ce3D (*Li et al., 2017*), in which the concentration of iohexol (Nycodenz, AN1002424, Accurate Chemical and Scientific Corp) is increased to match the refractive index (RI) of the mounting solution to that of standard microscopy oils (nD = 1.515; not utilized in this study) and 1-thioglycerol is omitted to reduce toxic compounds within the solution (thereby making the solution safer to handle) and increase shelf life. To account

for dilution by water from a sample (including agarose volume) we used Ce3D++ solution, in which the iohexol concentration is further increased. Ce3D++ was designed to produce desired RI after two incubations - detailed protocol is available upon request. The protocol for preparing these solutions is as follows: in a 50 mL tube, add iohexol powder (20 g for Ce3D+ or 20.83 g for Ce3D++), then 10.5 g of 40% v/v solution of N-Methylacetamide (M26305, Sigma) in 1X PBS, and then 22.5 mg Triton-X-100 (T8787, Sigma) for a final concentration of 0.1%. The solution is mixed overnight on an orbital rocker then stored at room temperature. Embedded embryos were cleared using 2 changes of Ce3D++ solution while rocking at room temperature for a twenty-four-hour period.

## Imaging

All images were obtained on an Olympus FV1000 confocal microscope, with an image size of 1620 × 1200 pixels and with a Kalman averaging of 3 frames. To capture the entire tissue a 10x UPlanApo objective (NA = 0.4) was used achieving a pixel size of 0.9 x 0.9 µm. Tissues were oriented in the same way with the anterior-posterior axis of the tissue oriented from left-right within the center of the field. Microscope settings were kept consistent between imaging. Intensity calibration and shading correction was performed as previously described (*Model, 2006*), however fluorophore dye concentrations used for shading correction were 0.05 mg/L for fluorescein (Sigma 46960), 1 mg/L for acid blue 9 (TCI CI42090), and 2 mg/L for rose bengal (Sigma 198250). Dyes were placed in coverslip bottomed dishes. The Shading Correction plugin within Fiji was then used with the median flat field images acquired using the dye solutions.

## Image processing

Images within figures were processed using Fiji (*Schindelin et al., 2012*) and represent max projections of z-stacks. Compared images are presented with identical intensity ranges for each channel. Orthogonal projections were made using Imaris software (Imaris V9.2.1, Bitplane Inc).

## Statistical analysis

For all analysis at least three embryos were used unless otherwise stated in the text or caption. Significance was determined using a Students two-tailed t-test.

## Imaris fluorescence quantification and modeling

### HCR data

Flat-fielded image stacks were imported from Fiji into Imaris (Imaris V9.2.1, Bitplane Inc). A baseline subtraction was then performed for each probe, using a cutoff value specific for each probe and fluorophore combination. For all probes, except *Hes7*, the baseline cutoff was adjusted until signal from a tissue known not to express the gene was no longer detectable (e.g. neural epithelium for *Tbx6*; *Nowotschin et al., 2012*; *Chapman et al., 1996*). For *Hes7*, this cutoff was determined by using embryos that were homozygous for a *Hes7* null allele (*Bessho, 2001*) that lacked sequences complementary to the HCR probes; the cutoff was chosen that resulted in no signal in these mutants (*Figure 5—figure supplement 5*).

The Surface model tool was used to build surfaces for each expression domain to be quantified except for *Fgf8* and *Hes7*, which were quantified using a surface derived from the *Tbx6* expression domain. Quantification of *Spry2*, *Spry4* and *Etv4* were determined by measurement of fluorescence intensity per cubic micrometer within the volume the respective gene expression domain. Surface models were generated using a surface detail value of 3 µM, absolute intensity setting, and an absolute intensity threshold cutoff that was set to exclude background signal in tissues known not to express the gene being modeled. Once established for each probe, the same intensity threshold cutoff values were used for generating surfaces for all embryos. For generating *Tbx6* surfaces, a range of absolute intensity threshold cutoffs were used to achieve a final surface volume between 9.0e6 and 1.1e7 µm$^3$. Values for mRNA expression represent the intensity sum within the volumetric model divided by the volume of the model.

### Immunostaining data

In images where NICD was immunostained, the surface creation tool was used. The mesoderm was selected, eliminating the ectodermal and endodermal tissue, on alternating z-planes through the

entirety of the z-stack. The resulting pattern was interpolated for intervening planes. Values for NICD expression represent the intensity sum within the volumetric model divided by the volume of the model.

## Spot modeling

Spot diameter was set to 4 µM with a point spread function value of 8 µM, local background subtraction was used, and sum intensity of the channel being modeled was used to threshold. Threshold cutoff values for spot models were set at 1500 for all figures except *Figure 6*. In *Figure 6* the thresholds values (a.u.) for the sum intensity are: low (blue), 1,000–6,000; low-medium (green), 6,000–11,000; medium (yellow), 11,000–16,000; medium-high (red), 16,000–21,000; high (white), 21,000 or greater. 'Heat maps' in *Figures 4* and *7* were generated within the Imaris software using a linear color scale; the scale in *Figure 4* ranged from intensity values of 7000 (blue) to 20,000 (red), the scale in *Figure 7* ranged from 1000 (blue) to 15,000 (red).

## Acknowledgements

We thank Hector Escriva for helpful discussion. We also thank E Kamiya and W Heinz of the NCI Optical Microscopy and Image Analysis Lab, for their assistance with clearing using Ce3D+ and Imaris, respectively. We are grateful for technical assistance from C Bonatto, T Kuruppu, C Elder, E Truffer, and M Boylan. We thank M Kaltcheva, P Abete-Luzi and A Perantoni for critical reading of the manuscript.

## Additional information

### Funding

| Funder | Grant reference number | Author |
| --- | --- | --- |
| National Cancer Institute | ZIA BC010338-19 | Mark Lewandoski |

The funders had no role in study design, data collection and interpretation, or the decision to submit the work for publication.

### Author contributions

Matthew J Anderson, Conceptualization, Data curation, Formal analysis, Validation, Investigation, Visualization, Methodology, Writing - original draft, Writing - review and editing; Valentin Magidson, Ryoichiro Kageyama, Resources, Writing - review and editing; Mark Lewandoski, Conceptualization, Resources, Data curation, Formal analysis, Supervision, Funding acquisition, Investigation, Methodology, Writing - original draft, Project administration, Writing - review and editing

### Author ORCIDs

Matthew J Anderson https://orcid.org/0000-0001-9387-5743
Mark Lewandoski https://orcid.org/0000-0002-1066-3735

### Ethics

All work with mice was approved and monitored by the NCI-Frederick Animal Care and Use Committee (ACUC). Mice were grouply housed in AAALAC (Association for Assessment and Accreditation of Laboratory Animal Care International) accredited facilities and in accordance with ACUC guidelines (Animal Study Protocol 17–069).

### Decision letter and Author response

Decision letter https://doi.org/10.7554/eLife.55608.sa1
Author response https://doi.org/10.7554/eLife.55608.sa2

## Additional files

### Supplementary files
- Transparent reporting form

### Data availability
All data generated or analyzed during this study are included in the manuscript and supporting files.

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
