## [Decision Letter]

**Acceptance summary:**

In this paper, Anderson and colleagues provide evidence for a specific role of *Fgf4* in the control of *Hes7* oscillations in the mouse presomitic mesoderm (PSM). Using mouse mutants harboring conditional deletions of *Fgf4* or *Fgf8* in the PSM, they show that only *Fgf4* deletion leads to significant segmentation defects first visible in the oscillatory pattern of *Hes7* and NICD and in the stripes of Mesp2 ultimately leading to abnormal vertebrae. They propose and provide evidence for a model wherein *Fgf4* is required to maintain *Hes7* above a defined threshold to ensure proper negative regulation of its expression associated to oscillations. This is an important study that expands understanding of the mechanisms underlying the segmentation clock.

**Decision letter after peer review:**

Thank you for submitting your article "*Fgf4* is critical for maintaining *Hes7* levels and Notch oscillations in the somite segmentation clock" for consideration by *eLife*. Your article has been reviewed by Marianne Bronner as the Senior Editor, a Reviewing Editor, and three reviewers. The following individual involved in review of your submission has agreed to reveal their identity: Olivier Pourquié (Reviewer #1).

The reviewers have discussed the reviews with one another and the Reviewing Editor has drafted this decision to help you prepare a revised submission.

Summary:

In this paper, Anderson and colleagues provide evidence for a specific role of *Fgf4* but not *Fgf8* in the control of *Hes7* oscillations in the mouse presomitic mesoderm (PSM). Using mouse mutants harboring conditional deletions of *Fgf4* or *Fgf8* in the PSM, they show that only *Fgf4* deletion leads to significant segmentation defects first visible in the oscillatory pattern of *Hes7* and NICD and in the stripes of Mesp2 ultimately leading to abnormal vertebrae. Interestingly, these defects are restricted to the cervico-thoracic region. Remarkably, in these mutants, expression of PSM markers and FGF targets is normal indicating that *Fgf8* is sufficient to maintain the wavefront. They then use quantitative fluorescent in situ hybridization (HCR) to analyze the *Fgf4* mutants and show that while the periodic expression of *Hes7* is maintained, it is not as sharp as in controls, leading to abnormal Mesp2 expression at the wavefront. Importantly, they show a significant downregulation of *Hes7* expression in the PSM of mutants compared to controls and propose a model wherein *Fgf4* is required to maintain *Hes7* above a defined threshold to ensure proper negative regulation of its expression associated to oscillations. They provide further evidence for this model by showing that *Fgf4* (but not *Fgf8*) genetically interacts with *Hes7* such that double heterozygote mutants exhibit a strong increase of segmentation defects compared to single mutants.

Essential revisions:

1) *Fgf4* expression analysis should be performed by HCR in both wild type and mutant embryos. In the case of the WT embryos, it should include both E8.5 and E9.5. *Fgf4* expression does not seem to be well characterized and the absence of phenotype at later stages could just reflect absence of *Fgf4* expression at that level. This could also show the extension of *Fgf4* expression in the PSM. As pointed out by the reviewers, T-cre activity is not restricted to the PSM as it starts in the primitive streak. Checking *Fgf4* expression in the mutants will also show where and the extent of its inactivation.

2) Similarly, characterization of *Lfng* expression and better assessment of Mesp2 activation in the mutants is essential.

3) It would be nice if you could also perform dynamic analysis of cycling activity in the PSM of *Fgf4* mutants, if you have access to the reporter strain crossed with the T-cre-*Fgf4* line. Similarly, dynamic evaluation of the cycling activity in the PSM and/or solving the mechanism of *Hes7* regulation by *Fgf4* would be a nice addition, but we would understand if this might take too long under the present circumstances.

We include the full reviews below for your information.

Reviewer #1:

The role of FGF signaling in somite patterning has so far been difficult to understand as it appears to be involved in both the control of the oscillations and in the formation of a posterior gradient involved in controlling the positioning of the determination front. I find this a very interesting study which combines elegant mouse genetics coupled to novel quantitative methods of gene expression analysis in vivo to demonstrate the implication of *Fgf4* in the regulation of the oscillations. This is a novel finding that should be of interested to the community of developmental biologists and beyond. The paper is suitable for *eLife* provided the comments below are addressed.

Essential revisions:

One of puzzling observations of the work is the irregular pattern of NICD expression. It is unclear to me why the disruption of *Hes7* expression should affect NICD. One explanation could be through *Lfng* which is regulated by *Hes7* and which in turn regulates Notch signaling. I think analyzing *Lfng* expression in the *Fgf4* mutants would be important for this study to be able to conclude on this point.

Also, it would be interesting to examine Axin2 expression to see if the effect is strictly restricted to Notch targets.

Reviewer #2:

This paper describes the effect that removal of *Fgf4* from the presomitic mesoderm (PSM) of mouse embryos has on Notch cyclic activity in this region and axial skeletal development. The authors show that embryos in which *Fgf4* was conditionally inactivated in the PSM had vertebral malformations in the cervical and thoracic regions, compatible with abnormal segmentation in the paraxial mesoderm. This effect was not seen in similar *Fgf8* mutants, the other FGF molecule thought to be functionally relevant in this embryonic region. They then show abnormal cyclic behaviour of *Hes7* and *Lfng* in the PSM of *Fgf4* mutant embryos at E8.5 but not ~E9.5 (22-24 somites). Through quantitative estimations using whole mount in situ HCR the authors show that in *Fgf4* mutant embryos *Hes7* expression is overall reduced in the PSM. To explain the link between *Hes7* expression levels and abnormal Notch cyclic behaviour the authors propose that the reduced *Hes7* expression would interfere with the negative feedback loop on its own transcription, eventually impacting the cyclic Notch signalling activity in the PSM. The relevance of *Hes7* downstream *Fgf4* signalling was further shown by introducing the *Fgf4* mutation into the *Hes7* heterozygous background, which resulted in a clear increase in the strength of the skeletal phenotypes relative to those observed in *Fgf4* mutant or *Hes7* heterozygote embryos.

This work provides evidence of a connection between *Fgf4* signalling and the cyclic behaviour of Notch signalling in the PSM. It also represents the first direct quantitative description of the importance of precise expression levels of a particular component of the Notch pathway to guarantee proper function of the regulatory network leading to Notch cyclic signalling activity, and the impact that this actually has on paraxial segmentation. It also shows the value of HCR to understand the impact of quantitative parameters in gene function in vivo.

I have a few comments that I think could help increasing the impact of this manuscript in order to further understand the mechanisms of paraxial segmentation and the specificity in FGF signalling activity.

1) The current paradigm for somitogenesis includes a FGF signalling gradient in the PSM, acting as an essential component of clock and wavefront model. How does the data in this manuscript fit in this model? For instance, *Fgf8* has been considered the key FGF signal forming the relevant gradient, but *Fgf8* inactivation seems to have no negative impact on somitogenesis. However, *Fgf4* inactivation alone is able to interfere with proper somitogenesis. Does *Fgf4* generate a gradient in the PSM? This could be explored by whole mount in situ HCR. In case such gradient is not observed, or it does not cover the PSM length, is a FGF gradient required for somitogenesis?

2) In line with the previous comment, it looks like the *Fgf4* is required to set *Hes7* expression levels and that this alone has significant effects on the functional dynamics of Notch signalling, eventually affecting segmentation. This might actually fit better with a role of FGF signalling on the cells added by the axial progenitors to the posterior end of the PSM as the embryo grows, rather than with interference with a FGF gradient in the PSM. Indeed, an independent double *Fgf4*/*Fgf8* gene inactivation in the axial progenitor zone (Boulet and Capecchi, 2012) produced a phenotype more consistent with these signals modulating the production of new paraxial mesodermal tissue to be incorporated into the PSM as the embryos grow. The phenotype of the double T-cre-*Fgf8*/T-cre-*Fgf4* mutants mentioned in this manuscript could also be interpreted along the same lines.

3) Do the authors have an insight into the mechanism by which *Fgf4* modulates *Hes7* expression levels? Also, why is *Fgf8* not able to substitute for *Fgf4* in this function, when it is expressed at higher levels in the posterior embryo than *Fgf4*? This is important as it could also help understanding signalling specificity from the different FGF molecules (and they are too often used interchangeably in many experimental designs). Also, it questions the use of the analysis of *Fgf8* gradients to interpret segmentation defects in mutant embryos.

4) An interesting finding in this manuscript is that the effects of inactivating *Fgf4* are observed in cervical, thoracic and anterior lumbar regions but not in more posterior areas of the skeleton. This is reminiscent of what has been observed for *Lfng*: while its total inactivation affects paraxial segmentation throughout the anterior posterior axis (Zhang and Gridley, 1998; Evrard et al., 1998), *Lfng* cyclic activity seems to be required for presacral vertebrae but dispensable for posterior embryonic regions (Shifley et al., 2008). The authors interpret this differential *Fgf4* effect in terms of different speed of somitogenesis at the two axial levels. However, as that the anterior regions derive from the epiblast through the primitive streak, whereas more posterior regions derive from the tail bud, it is possible that somitogenesis itself could be mechanistically not so uniform and contain relevant differences in areas produced by progenitors in the epiblast or tail bud. Such differences could explain the differential phenotypes in anterior and posterior embryonic regions upon *Fgf4* inactivation or the different requirements for *Lfng* cyclic behaviour. Also, could differences in T-cre driver activity in the epiblast and tail bud play a role in the regional-specific phenotypes of the *Fgf4* mutation?

Reviewer #3:

In this manuscript the authors provide convincing genetic evidence for a role of *Fgf4* in controlling somite formation in mouse embryos. I find this work of high quality and importance. While in a previous study, the authors had concluded that "*Fgf4* [...] is not required for somitogenesis per se" (Page 4021 in Naiche et al., 2011) their current work clearly identifies *Fgf4* as an important player particularly for cervical somite formation. Moreover, they suggest a possible mechanism, i.e. the control of *Hes7* expression by *Fgf4* (see below my comments). These findings will stimulate follow up studies needed to clarify the precise mechanism of how *Fgf4* impacts segmentation, possible via an effect on segmentation clock oscillations. As such this will be an important contribution to the field.

My main concerns relate to the interpretation of the data and mechanism underlying the phenotype and also the reference to previous work.

1) Interestingly, *Fgf4* KO results in a regionally confined phenotype, i.e. mainly cervical and thoracic defects, while more posterior vertebrae are not affected. This is mirrored by a slight downregulation of *Hes7* only during early stages of somite formation and a slight mis-expression of Mesp2, again only in the early stages of somite formation.

As possible explanation the authors suggest that anterior somites, which are forming at a slightly faster rate, are more susceptible to the downregulation of Hes7. However, this interpretation seems to be difficult to reconcile with previous data, i.e. it has been shown that the phenotype of *Hes7* KO is LESS severe in more rostral somites and more severe in more posterior somites ( Bessho et al., 2001). If indeed the loss of *Fgf4* is mediated by the downregulation of Hes7, wouldn't one expect to find a similar bias, i.e. rostral vs. caudal, in phenotype severity in both conditions, i.e. *Fgf4* KO and *Hes7* KO? This point needs clarification.

2) Another possible mechanism compatible with the data presented is an issue with cell-to-cell synchronization upon *Fgf4* depletion. This is supported by the data on NICD expression, for instance, which is more "irregular" in the PSM. In turn, synchronization is known to rely on Notch-Δ interactions and both, Notch1 and Dll1 have been shown to exhibit oscillatory expression dynamics.

Therefore, another analysis that I think is urgently missing is a quantification of Dll1 and Notch1 (mRNA and/or protein) in *Fgf4* mutants. If affected, this could provide a mechanistic insight into a potential loss of synchrony between cells in *Fgf4* mutants.

3) At the conceptual level, the role suggested for *Fgf4* is to control segmentation clock oscillations, not the wavefront. The authors state: "neither a change in wavefront activity nor a change in determination front position explains the aberrant pattern of Mesp2 expression in *Fgf4* mutants."(subsection “Wavefront gene expression is normal in *Fgf4”*). What remains unclear, unfortunately, is what the effect of *Fgf4* on the segmentation clock oscillations actually is and how it mechanistically works. Hence, while the key conclusion is 'Together, these data suggest the aberrant Mesp2 patterning *Fgf4* mutants can be explained by imprecise oscillations of Notch signaling." (subsection “Oscillation of Notch family components is altered in the PSM of *Fgf4* mutants”), there is no direct data on oscillation dynamics. In my view, the static analysis of clock phases (Figure 5) is difficult to interpret. Real-time imaging quantification are needed, in my view, in particular if relevant for the main conclusion(s) of the study, as in this case (see title of the manuscript!). I would advise to make this extra effort and quantify segmentation oscillations in real-time and reveal, whether indeed "imprecise oscillations" are found. This would substantially strengthen this manuscript. If deemed "beyond the scope of this paper...", I would argue that the conclusions regarding the effect of Fg4 on the segmentation clock need to be rephrased and adjusted accordingly, i.e. the tile would need to be rephrased, in my view.

4) "We demonstrate that, while *Fgf8* is not required for somitogenesis, *Fgf4*

is required for normal Notch oscillations and patterning of the vertebral column (Introduction)." As stated above, this contrasts with the previous conclusion by the authors, i.e. "that *Fgf4* likewise is not required for somitogenesis per se (Page 4021 in Naiche et al., 2011). Given that the similar read-out was used at least partially, i.e. skeletal preparations, there needs to be a reference and explanation to avoid confusion of the reader.

5) Axial level of effects – somites vs. vertebrae:

The comparison between effects at the level of segmentation of cranial somites vs. cervical vertebrae needs to be clarified. Figure 1E indicates that the first defects are found at vertebrae C5, which actually is formed by somite 10/11. However, somite defects are visible much earlier, i.e. starting at somite 3 (Figure 1J). I think this needs clarification or at least commenting. In this regard, it is again interesting to relate to the seminal *Hes7* KO report (Beshho et al., 2011), as in this context, initial irregularities at the level of somite segmentation were found to be corrected during later development. Is a similar explanation appropriate here? This could also mean that a relevant score (in Figure 1J) could include how severe the phenotype is at the level of Mesp2 expression, i.e. extend of irregularity, for instance. The idea here could be that while minor Mesp2 irregularities might be corrected, more extended Mesp2 irregularities cannot and lead to a phenotype. Such a Mesp2 severity scoring could hence provide additional mechanistic clues into the observed phenotype.

6) One key remaining question is the mechanism of *Fgf4* function. I think that the evidence to claim " that *FGF4* controls Notch pathway oscillations through the transcriptional repressor, *HES7*" (Abstract) is not convincing. While indeed *Hes7* expression appears irregular in *Fgf4* mutants, this could also be interpreted as an indirect consequence of *Fgf4* depletion, for instance as consequence of synchronization issues (see above). The finding that *Fgf4* and *Hes7* genetically synergize does not provide evidence that *Hes7* downregulation is a direct consequence of *Fgf4* depletion, either. In this regard, it is of note that *Hes7* gene expression regulation has been intensively studied and it was shown that the *Hes7* enhancer is controlled (and bound) by Tbx6, Mesogenin and Nicd (Hayashi et al., 2018). What is the evidence *Fgf4* and FGF signaling controls *Hes7* expression directly? I suggest this uncertainty in interpretation is better reflected in the phrasing of the conclusion (again, see Title) and interpretations until the mechanism of *Fgf4* is revealed.

---

## [Author Response]

Essential revisions:1) Fgf4 expression analysis should be performed by HCR in both wild type and mutant embryos. In the case of the WT embryos, it should include both E8.5 and E9.5. Fgf4 expression does not seem to be well characterized and the absence of phenotype at later stages could just reflect absence of Fgf4 expression at that level. This could also show the extension of Fgf4 expression in the PSM. As pointed out by the reviewers, T-cre activity is not restricted to the PSM as it starts in the primitive streak. Checking Fgf4 expression in the mutants will also show where and the extent of its inactivation.

We think of this critique as asking 3 questions about *Fgf4* expression and function as follows.

1.1) Characterize PSM-specific Fgf4 expression more thoroughly. Specifically, “the absence of phenotype at later stages could just reflect absence of Fgf4 expression at that level.”? (This relates to reviewer 2, comment 4).

In the first draft of this manuscript, Figure 1A, A’ and B showed *Fgf4* expression at somite stages 5 and 10 in wild-type embryos, when defects occur in *Fgf4* mutants. We have now added Figure 1—figure supplement 3, which shows PSM-specific *Fgf4* expression at somite stages 6 and 24 (via HCR) (Panels A, A’, C, C’) as well as E9.5 through 12.5 in the tailbud (via WISH) (panels E-G’). The somites and vertebrae generated at all of these stages (except somite stage 6) are normal, thus directly answering this question. The following new text (subsection “FGF4 activity is required in the presomitic mesoderm for normal segmentation of rostral somites”) addresses this question:

“We considered whether this normal *Mesp2* expression (Figure 1—figure supplement 2) and the resulting proper somite patterning caudal to somite 22 that occurs in *Fgf4* mutants (Figure 1J), might reflect an absence of PSM-specific *Fgf4* expression in normal development at later stages. However, we detected *Fgf4* expression throughout the PSM at somite stage 22 through E12.5 (Figure 1—figure supplement 3C, C’, E-G’).”

1.2) Show the efficiency of TCre-mediated inactivation of *Fgf4*. “As pointed out by the reviewers, T-cre activity is not restricted to the PSM as it starts in the primitive streak. Checking Fgf4 expression in the mutants will also show where and the extent of its inactivation.” (related to reviewer 2, comment 4). In the new Figure 1—figure supplement 3, we performed an HCR analysis of *Fgf4* at somite stages 6 and 24, using a probe for sequences deleted by Cre (therefore we are screening Cre deletion of *Fgf4*). This shows *Fgf4* is deleted both in the PSM and primitive streak.

The following text (subsection “FGF4 activity is required in the presomitic mesoderm for normal segmentation of rostral somites”) describes this new data:

“We then considered whether TCre-mediated inactivation of *Fgf4* resulted in loss of *Fgf4* expression in the PSM throughout development. We expected this would be the case because TCre is active in the primitive streak at E7.5^36^ and, in prior studies, has been shown to inactivate *Fgf4* and *Fgf8* in the primitive streak and PSM, prior to formation of the first somite ^28,36^. Indeed, we detect no *Fgf4* expression in the PSM at somite stage 6 (Figure 1—figure supplement 3B, B’), when aberrant somites form, nor at somite stage 24 (Figure 1—figure supplement 3D, D’), when normal somites form. Therefore, we conclude that *Fgf4* is expressed within the PSM throughout development, but is required for normal somitogenesis only at early stages. When *Fgf4* is inactivated early, in the caudal embryo, abnormally patterned *Mesp2* expression occurs only at these early stages, causing the observed vertebral defects in *Fgf4* mutants.”

Also, we edited the Discussion to read “We describe mutants with only *Fgf4* or *Fgf8* inactivated specifically in the primitive streak and PSM.”

1.3) “This could also show the extension of Fgf4 expression in the PSM.”

We think this relates to reviewer 2, comment 1, asking whether *Fgf4* is expressed in a gradient. In the caption to Figure 1—figure supplement 3, we make the observation “*Fgf4* does not appear to be expressed in a gradient in the normal PSM” Also, please see our response to reviewer 2, comment 1 regarding the idea of gradients.

2) Similarly, characterization of Lfng expression and better assessment of Mesp2 activation in the mutants is essential.

The *Lfng* suggestion relates to reviewer 1, comment 1. We have now added Figure 4—figure supplement 2, which is a more detailed analysis of *Lfng* expression in mutants and littermate controls. We describe this new data (subsection “Oscillation of Notch family components is altered in the PSM of *Fgf4* mutants”) with the following text.

“A synchronous oscillatory NICD pattern is achieved through negative feedback loops controlled by *Lfng*^15^and *Hes7*^14^. *Lfng* encodes a glycosyltransferase that represses Notch signaling in the PSM ^11,44,45^, possibly via glycosylation of DLL1 and/or DLL3 ^46,47^. *Hes7* is a basic helix-loop-helix transcription factor that is essential for repression of a number of Notch target genes ^32,48,49^. *Hes7* and *Lfng* are themselves targets of activated Notch and therefore their expression oscillates ^14^. Disruption of either gene causes uncoordinated Notch signaling in the PSM ^14^.

Although oscillatory expression of *Hes7* and *Lfng* are apparent in both *Fgf4* mutants and littermate controls (Figure 4D-E’’), the mutant patterns are asynchronous and indistinct, similar to the NICD mutant pattern (Figure 4D-E’’). *Lfng* expression is particularly disordered in the anterior PSM (Figure 4E’ and E’’; Figure 4—figure supplement 1), where both NICD and *Mesp2* also show aberrant patterning (Figure 4C, C’ and Figures 1L, Figure 2B). *Hes7* expression is likewise disordered in the mutant PSM at these stages (Figure 4E, E’’). Together, these data suggest the aberrant *Mesp2* pattern in *Fgf4* mutants can be explained by asynchronous oscillations of Notch signaling.”

Also in subsection “*Hes7* expression is reduced in the PSM of *Fgf4* mutants”:

“Moreover, HES7 directly represses *Lfng* transcription ^49^and is required for normal *Lfng* oscillations within the PSM ^11,44,45,48^. Therefore, defective *Hes7* expression could account for the abnormal NICD and *Mesp2* patterning in *Fgf4* mutants, possibly by causing aberrations in *Lfng* expression.”

Regarding a “*better assessment of Mesp2”,* this is linked to reviewer 3, comment 5, which argues for an *Mesp2* severity scoring data set. The reviewer suggested that *Mesp2* irregularities might be relatively minor in somites 1-5, when altered somite patterning occurs (Figure 1J) but doesn’t lead to vertebral defects as frequently (Figure 1E) as more caudal somites.

However, reviewer 3 also makes the excellent point that in Bessho et al., *Hes7* null mutants, show “initial irregularities at the level of somite segmentation were found to be corrected during later development”. Indeed, we think this observation relates to the *Fgf4* mutant- and describe the resulting changes below in our response to reviewer 3, comment 5.

3) It would be nice if you could also perform dynamic analysis of cycling activity in the PSM of Fgf4 mutants, if you have access to the reporter strain crossed with the T-cre-Fgf4 line. Similarly, dynamic evaluation of the cycling activity in the PSM and/or solving the mechanism of Hes7 regulation by Fgf4 would be a nice addition, but we would understand if this might take too long under the present circumstances.

This relates to reviewer 2, comment 3 and reviewer 3, comments 3 and 6. We agree that dynamic analysis of cycling activity would be ideal. One of our co-authors (Dr Kageyama, Kyoto University, Kyoto, Japan) sent us mice carrying the Hes7-Luc transgene (Takashima et al., 2011). We then generated the appropriate crosses to analyze this Hes7-Luc transgenic luciferase activity live in *Fgf4* mutants. Unfortunately, we found the luciferase activity too weak at the necessary early somite stages, particularly in *Fgf4* mutants, which have a ~ 50% reduction in both endogenous and Hes7-Luc expression, to obtain data of sufficient quality for analysis.

We are currently importing a more sensitive fluorescent Hes7 transgenic reporter mouse line, also generated by Dr. Kagayama’s group, called “Hes7-Achilles” (Yoshioka-Kobayashi et al., 2020). We expect this reporter to overcome these impediments, but the necessary importation, genetic crosses and experimentation are easily a year or more in the future. We refer to this reporter (in a different context) when discussing future plans (Discussion section). “In future experiments, we will test this hypothesis directly by measuring the clock rate, using recently generated fluorescent transgenic reporters of *Hes7*^73^.”

Likewise, we are planning experiments to probe the mechanism of *Hes7* regulation by FGF4. We have happy to consider submitting these results as a Research Advance in *eLife*, or as a full-fledged separate e*Life* publication (if warranted) per recent e*Life* policy regarding “Publishing in the time of Covid-19” (https://elifesciences.org/articles/57162 section 1).

Reviewer #1:The role of FGF signaling in somite patterning has so far been difficult to understand as it appears to be involved in both the control of the oscillations and in the formation of a posterior gradient involved in controlling the positioning of the determination front. I find this a very interesting study which combines elegant mouse genetics coupled to novel quantitative methods of gene expression analysis in vivo to demonstrate the implication of Fgf4 in the regulation of the oscillations. This is a novel finding that should be of interested to the community of developmental biologists and beyond. The paper is suitable for eLife provided the comments below are addressed.

We thank the reviewer for his opinion:

Essential revisions:One of puzzling observations of the work is the irregular pattern of NICD expression. It is unclear to me why the disruption of Hes7 expression should affect NICD. One explanation could be through Lfng which is regulated by Hes7 and which in turn regulates Notch signaling. I think analyzing Lfng expression in the Fgf4 mutants would be important for this study to be able to conclude on this point.

This is a very helpful criticism. Although we had *Lfng* expression data in the original draft (Figure 4 D’,- E”), we didn’t connect it as explicitly as we should have to the reduced *Hes7* levels, nor to the altered NICD pattern. We have now added Figure 4—figure supplement 1 and textual changes, detailed in the response to Essential revisions 2 above.

Also, it would be interesting to examine Axin2 expression to see if the effect is strictly restricted to Notch targets.

We agree that it would be very interesting to look at WNT signaling, given that WNT and FGF signaling are known to interact in the PSM – and that *Axin2* expression would be a good first start. However, Axin2 (while a good WNT reporter) is not required for somitogenesis (Yu et al., 2005). Therefore, any observed changes require further studies to determine whether WNT signaling is acting epistatically between *Fgf4* and *Hes7* or just downstream of *Fgf4*, in parallel to *Fgf4*’s effect on *Hes7*. We will perform such experiments in the future; we propose that the current study supports our conclusions, focusing an FGF4 requirement for *Hes7* expression. Also, in a new paragraph in the Discussion sectiion includes the sentence “Additionally, it is possible that FGF4 signaling is interfacing with WNT signaling, which also has been shown to regulate *Hes7*^6529,30^.”

Reviewer #2:This paper describes the effect that removal of Fgf4 from the presomitic mesoderm (PSM) of mouse embryos has on Notch cyclic activity in this region and axial skeletal development. The authors show that embryos in which Fgf4 was conditionally inactivated in the PSM had vertebral malformations in the cervical and thoracic regions, compatible with abnormal segmentation in the paraxial mesoderm. This effect was not seen in similar Fgf8 mutants, the other FGF molecule thought to be functionally relevant in this embryonic region. They then show abnormal cyclic behaviour of Hes7 and Lfng in the PSM of Fgf4 mutant embryos at E8.5 but not ~E9.5 (22-24 somites). Through quantitative estimations using whole mount in situ HCR the authors show that in Fgf4 mutant embryos Hes7 expression is overall reduced in the PSM. To explain the link between Hes7 expression levels and abnormal Notch cyclic behaviour the authors propose that the reduced Hes7 expression would interfere with the negative feedback loop on its own transcription, eventually impacting the cyclic Notch signalling activity in the PSM. The relevance of Hes7 downstream Fgf4 signalling was further shown by introducing the Fgf4 mutation into the Hes7 heterozygous background, which resulted in a clear increase in the strength of the skeletal phenotypes relative to those observed in Fgf4 mutant or Hes7 heterozygote embryos.This work provides evidence of a connection between Fgf4 signalling and the cyclic behaviour of Notch signalling in the PSM. It also represents the first direct quantitative description of the importance of precise expression levels of a particular component of the Notch pathway to guarantee proper function of the regulatory network leading to Notch cyclic signalling activity, and the impact that this actually has on paraxial segmentation. It also shows the value of HCR to understand the impact of quantitative parameters in gene function in vivo.I have a few comments that I think could help increasing the impact of this manuscript in order to further understand the mechanisms of paraxial segmentation and the specificity in FGF signalling activity.

We thank the reviewer for his/her opinion: This work provides evidence of a connection between Fgf4 signalling and the cyclic behaviour of Notch signalling in the PSM. It also represents the first direct quantitative description of the importance of precise expression levels of a particular component of the Notch pathway to guarantee proper function of the regulatory network leading to Notch cyclic signalling activity, and the impact that this actually has on paraxial segmentation. It also shows the value of HCR to understand the impact of quantitative parameters in gene function in vivo.

1) The current paradigm for somitogenesis includes a FGF signalling gradient in the PSM, acting as an essential component of clock and wavefront model. How does the data in this manuscript fit in this model? For instance, Fgf8 has been considered the key FGF signal forming the relevant gradient, but Fgf8 inactivation seems to have no negative impact on somitogenesis. However, Fgf4 inactivation alone is able to interfere with proper somitogenesis. Does Fgf4 generate a gradient in the PSM? This could be explored by whole mount in situ HCR. In case such gradient is not observed, or it does not cover the PSM length, is a FGF gradient required for somitogenesis?

Regarding a role of an FGF signaling gradient in somitogenesis. We address the question of *Fgf4* expression and a potential gradient in Essential revisions 1.3 above. We enthusiastically agree with the reviewer that the subject of FGF gradients in the PSM is an important and fascinating one. However, we suggest that a deep analysis of FGF gradients via HCR analysis, including an experimental investigation into their functionality, is beyond the scope of the current paper.

2) In line with the previous comment, it looks like the Fgf4 is required to set Hes7 expression levels and that this alone has significant effects on the functional dynamics of Notch signalling, eventually affecting segmentation. This might actually fit better with a role of FGF signalling on the cells added by the axial progenitors to the posterior end of the PSM as the embryo grows, rather than with interference with a FGF gradient in the PSM. Indeed, an independent double Fgf4/Fgf8 gene inactivation in the axial progenitor zone (Boulet and Capecchi, 2012) produced a phenotype more consistent with these signals modulating the production of new paraxial mesodermal tissue to be incorporated into the PSM as the embryos grow. The phenotype of the double T-cre-Fgf8/T-cre-Fgf4 mutants mentioned in this manuscript could also be interpreted along the same lines.

The caption of a new Figure 1—figure supplement 3, makes clear that *Fgf4* does not appear to be expressed in a gradient in the PSM (although note, we have only analyzed mRNA, not protein. Again, much more experimentation is needed, beyond the scope of the current study). We contend that FGF4 doesn’t affect the number of cells added by the axial progenitors to the posterior end of the PSM as the embryo grows, because there is no axial truncations in these *Fgf4* mutants (Anderson, 2016). Furthermore, a measure of the wavefront size (Figure 2) shows no difference at a stage when mispatterned somites are generated. In the two studies the reviewer references, where both *Fgf4* and *Fgf8* are inactivated, (Boulet and Capecchi, 2012; Naiche et al., 2011) such a truncation indeed occurs (although in Figure 4 of Naiche et al., we also demonstrated premature PSM differentiation). In the current draft we state this in subsection “Wavefront gene expression is normal in *Fgf4* mutants”, “Therefore, we conclude that wavefront activity is normal in *Fgf4* mutants, an insight consistent with the observation that normal axis extension occurs in these mutants, with no loss of caudal vertebrae ^42^.”

3) Do the authors have an insight into the mechanism by which Fgf4 modulates Hes7 expression levels? Also, why is Fgf8 not able to substitute for Fgf4 in this function, when it is expressed at higher levels in the posterior embryo than Fgf4? This is important as it could also help understanding signalling specificity from the different FGF molecules (and they are too often used interchangeably in many experimental designs). Also, it questions the use of the analysis of Fgf8 gradients to interpret segmentation defects in mutant embryos.

This is an intriguing and excellent question. (although we would point out that while the *Fgf8* mRNA pattern may be larger than that of *Fgf4*, we don’t know that this results in higher signaling levels). As mentioned above in Essential revision 3 and below in response to reviewer 3, comment 3 and 6, our future work will focus on how FGF4 regulates *Hes7*. Regarding Fgf8, we have added text to our Discussion section:

“This study and past work indicate that FGF4 and FGF8 have both shared and unique roles in the vertebrate embryo’s posterior growth zone. They redundantly encode wavefront activities, preventing PSM differentiation ^28,74^. In this capacity, when signaling together, they activate downstream canonical FGF target genes, such as *Spry2*, *Spry4* and *Etv4*^28^. Here, we show that the role that FGF4 signals play in maintaining *Hes7* mRNA levels cannot be replaced by FGF8.

Future efforts will explore how the signaling machinery within the PSM discriminates between these two FGF ligands. For example, our work argues for a reassessment of FGFR function in the PSM; prior work, in which the same Cre driver used here (TCre) inacativates *Fgfr1*^31^, thought to be the only FGF receptor gene expressed in the PSM, does not phenocopy the defects due to TCre-mediated loss of either *Fgf4* (this study) or *Fgf4/Fgf8*^28^.”

4) An interesting finding in this manuscript is that the effects of inactivating Fgf4 are observed in cervical, thoracic and anterior lumbar regions but not in more posterior areas of the skeleton. This is reminiscent of what has been observed for Lfng: while its total inactivation affects paraxial segmentation throughout the anterior posterior axis (Zhang and Gridley, 1998; Evrard et al., 1998), Lfng cyclic activity seems to be required for presacral vertebrae but dispensable for posterior embryonic regions (Shifley et al., 2008). The authors interpret this differential Fgf4 effect in terms of different speed of somitogenesis at the two axial levels. However, as that the anterior regions derive from the epiblast through the primitive streak, whereas more posterior regions derive from the tail bud, it is possible that somitogenesis itself could be mechanistically not so uniform and contain relevant differences in areas produced by progenitors in the epiblast or tail bud. Such differences could explain the differential phenotypes in anterior and posterior embryonic regions upon Fgf4 inactivation or the different requirements for Lfng cyclic behaviour. Also, could differences in T-cre driver activity in the epiblast and tail bud play a role in the regional-specific phenotypes of the Fgf4 mutation?

We agree that different axial levels have different requirements signaling and functional requirements eg, as shown in this paper and in (Shifley et al., 2008) as the reviewer points out. However, the reviewer is suggesting the transition from primary to secondary axis extension may correlate with the requirements for Fgf4 activity. We considered this intriguing idea, but do not think it helpful – because somites 22-32, which are still generated in the primary axis (i.e. rostral to neuropore closure) are unaffected in *Fgf4* mutants. Therefore, while all affected somites are in the primary axis, the caudal primary axis is not affected.

Regarding the TCre driver, our published work suggest inactivation of *Fgf4* and *Fgf8* prior to somitogenesis in all progenitors (see Figure 2 D,E in Perantoni et al., 2005 and Figures 1E, G, I and Figure 1—figure supplement 1D in Naiche et al., 2011). We have now added Figure 1—figure supplement 3, supporting these suppositions, as described above in Essential revision 1.2.

Reviewer #3:Summary:In this manuscript the authors provide convincing genetic evidence for a role of Fgf4 in controlling somite formation in mouse embryos. I find this work of high quality and importance. While in a previous study, the authors had concluded that "Fgf4 [...] is not required for somitogenesis per se" (Page 4021 in Naiche et al., 2011) their current work clearly identifies Fgf4 as an important player particularly for cervical somite formation. Moreover, they suggest a possible mechanism, i.e. the control of Hes7 expression by Fgf4 (see below my comments). These findings will stimulate follow up studies needed to clarify the precise mechanism of how Fgf4 impacts segmentation, possible via an effect on segmentation clock oscillations. As such this will be an important contribution to the field.My main concerns relate to the interpretation of the data and mechanism underlying the phenotype and also the reference to previous work.

We thank the reviewer for her/his opinion. These findings will stimulate follow up studies needed to clarify the precise mechanism of how Fgf4 impacts segmentation, possible via an effect on segmentation clock oscillations. As such this will be an important contribution to the field.

1) Interestingly, Fgf4 KO results in a regionally confined phenotype, i.e. mainly cervical and thoracic defects, while more posterior vertebrae are not affected. This is mirrored by a slight downregulation of Hes7 only during early stages of somite formation and a slight mis-expression of Mesp2, again only in the early stages of somite formation.As possible explanation the authors suggest that anterior somites, which are forming at a slightly faster rate, are more susceptible to the downregulation of Hes7. However, this interpretation seems to be difficult to reconcile with previous data, i.e. it has been shown that the phenotype of Hes7 KO is LESS severe in more rostral somites and more severe in more posterior somites ( Bessho et al., 2001). If indeed the loss of Fgf4 is mediated by the downregulation of Hes7, wouldn't one expect to find a similar bias, i.e. rostral vs. caudal, in phenotype severity in both conditions, i.e. Fgf4 KO and Hes7 KO? This point needs clarification.

We respectively disagree that the 50% reduction in *Hes7* mRNA (Figure 5H, Figure 5—figure supplement 4 and Figure 7E) is considered “slight”; it is both statistically significant and is a greater difference than that observed in *Hes7* heterozygotes (Figure 7E).

We suggest that a direct comparison of the conditional *Fgf4* mutants and *Hes7* null homozygotes phenotypes is a little complicated because, in *Fgf4* mutants, *Hes7* levels are reduced about 50% only at early stages (somite stage 5 – 6, Figure 5H and Figure 5—figure supplement 4) and then are found at normal levels later (somite stage 24-26, Figure 5 —figure supplement 5). Therefore, it is logical that more caudal regions of the vertebral column are unaffected in *Fgf4* mutants, unlike *Hes7* mutants. We make this more explicit in the Discussion section:

“We observed that in *Fgf4* mutants *Hes7* mRNA levels are reduced during rostral somitogenesis at E8.5 (5-6 somite stages), when defective somitogenesis occurs. At about E9.5 (24-26 somite stages), *Hes7* levels are unaffected, and the *Fgf4* mutant embryo is beginning to generate correctly patterned somites that will differentiate into normally patterned vertebrae.”

2) Another possible mechanism compatible with the data presented is an issue with cell-to-cell synchronization upon Fgf4 depletion. This is supported by the data on NICD expression, for instance, which is more "irregular" in the PSM. In turn, synchronization is known to rely on Notch-Δ interactions and both, Notch1 and Dll1 have been shown to exhibit oscillatory expression dynamics.Therefore, another analysis that I think is urgently missing is a quantification of Dll1 and Notch1 (mRNA and/or protein) in Fgf4 mutants. If affected, this could provide a mechanistic insight into a potential loss of synchrony between cells in Fgf4 mutants.

We agree that quantification of *Dll1* and *Notch* 1 would be an important step along the pathway of solving the mechanism of *Hes7* regulation by FGF4. But we contend that because NICD levels in *Fgf4* mutants are not significantly affected (Figure 4A), *Dll1* and *Notch1* levels are unlikely to be significantly reduced. (For example, *Dll1* and *Notch1* heterozygotes have a much less severe defect, see Figure 2 in Sparrow et al., 2012). However, we speculate that a careful Imaris-based HCR analysis may show an asynchronous expression pattern of these genes, as we have found for NICD, *Hes7* and *Lfng* (Figure 4, Figure 5, Figure 6, Figure 7). We agree with the reviewer that such an observation would support defective synchronization of oscillation (see point 3.3, below). However, we suggest this would be a step toward defining mechanism, but would be an incomplete analysis. Such experiments will be part of our mechanistic studies (see Essential revision 3, above).

3) At the conceptual level, the role suggested for Fgf4 is to control segmentation clock oscillations, not the wavefront. The authors state: "neither a change in wavefront activity nor a change in determination front position explains the aberrant pattern of Mesp2 expression in Fgf4 mutants."(subsection “Wavefront gene expression is normal in Fgf4”). What remains unclear, unfortunately, is what the effect of Fgf4 on the segmentation clock oscillations actually is and how it mechanistically works. Hence, while the key conclusion is 'Together, these data suggest the aberrant Mesp2 patterning Fgf4 mutants can be explained by imprecise oscillations of Notch signaling." (subsection “Oscillation of Notch family components is altered in the PSM of Fgf4 mutants”), there is no direct data on oscillation dynamics. In my view, the static analysis of clock phases (Figure 5) is difficult to interpret. Real-time imaging quantification are needed, in my view, in particular if relevant for the main conclusion(s) of the study, as in this case (see title of the manuscript!). I would advise to make this extra effort and quantify segmentation oscillations in real-time and reveal, whether indeed "imprecise oscillations" are found. This would substantially strengthen this manuscript. If deemed "beyond the scope of this paper...", I would argue that the conclusions regarding the effect of Fg4 on the segmentation clock need to be rephrased and adjusted accordingly, i.e. the tile would need to be rephrased, in my view.

The reviewer makes an excellent point. Our response to Essential revision 3, above, describes our efforts at live-imaging and mechanism as well as plans for moving forward…. With the reviewer’s criticism in mind, we edited the manuscript throughout to reflect the idea that oscillation synchrony is clearly affected (not listed here, but these should be evident in the “tracked” draft.). For example, we have edited the Title as suggested.

4) " We demonstrate that, while Fgf8 is not required for somitogenesis, Fgf4is required for normal Notch oscillations and patterning of the vertebral column (Introduction)." As stated above, this contrasts with the previous conclusion by the authors, i.e. "that Fgf4 likewise is not required for somitogenesis per se (Page 4021 in Naiche et al., 2011). Given that the similar read-out was used at least partially, i.e. skeletal preparations, there needs to be a reference and explanation to avoid confusion of the reader.

We apologize for the confusion. In Naiche et al., we used a broader definition of somitogenesis, equivalent to “somites form.” In the current manuscript we mean somitogenesis to mean “properly patterned somites”. To clarify this, we added a sentence in the beginning of the Discussion section:

“In previous work, we reported that somites form in such mutants ^28,36^; here we examine the patterning of these somites, the Notch pathway oscillations that control their formation, as well as the vertebral elements that they eventually form.”

5) Axial level of effects – somites vs. vertebrae:The comparison between effects at the level of segmentation of cranial somites vs. cervical vertebrae needs to be clarified. Figure 1E indicates that the first defects are found at vertebrae C5, which actually is formed by somite 10/11. However, somite defects are visible much earlier, i.e. starting at somite 3 (Figure 1J). I think this needs clarification or at least commenting. In this regard, it is again interesting to relate to the seminal Hes7 KO report (Beshho et al., 2011), as in this context, initial irregularities at the level of somite segmentation were found to be corrected during later development. Is a similar explanation appropriate here? This could also mean that a relevant score (in Figure 1J) could include how severe the phenotype is at the level of Mesp2 expression, i.e. extend of irregularity, for instance. The idea here could be that while minor Mesp2 irregularities might be corrected, more extended Mesp2 irregularities cannot and lead to a phenotype. Such a Mesp2 severity scoring could hence provide additional mechanistic clues into the observed phenotype.

We thank the reviewer for these excellent suggestions. We scored the severity of mutant *Mesp2* expression, and found no significant difference comparing somite stages 5-7 to somite stages 10-12. However, the reviewer’s instruction to look more closely at Beshho et al., 2001 was an excellent idea. In our original scoring of *Uncx4.1* patterning (Figure 1J), somites 1-6 were scored in somite stage 7 *Fgf4* mutants and somites 7-26 were scored in somite stage 26-30 *Fgf4* mutants. By examining the fifth and sixth somites in each of these two sets, we indeed found evidence that mispatterning of these somites corrected as the embryos developed, as the reviewer suggested, and as was reported in the *Hes7* KO (Beshho et al., 2001). Therefore, we replaced Figure 1J with only the data from the latter stage mutants and now show the more rostral scoring (of both sets of mutants) in Figure 1—figure supplement 1.

We now describe these analyses in subsection “FGF4 activity is required in the presomitic mesoderm for normal segmentation of rostral somites”:

“We examined gross somite patterning in *Fgf4* mutants by staining embryos at various stages for *Uncx4.1* mRNA, which marks the posterior somite compartment ^37,38^. Analysis of 19 E9.5 *Fgf4* mutants revealed a frequency of irregular *Uncx4.1* expression specifically in future cervical and thoracic somites with full penetrance (Figure 1F-J). We did not score the most rostral somites in these samples because *Uncx4.1* expression is absent in these early somites due to the differentiation of somites 1 – 4 to form the occipital bones ^39^. To examine these rostral somites we scored 20 *Fgf4* mutants at somite stage 7 and found defective *Uncx4.1* patterning beginning at somite 3 in this sample set (black bars, Figure 1—figure supplement 1). Interestingly, a majority of *Uncx4.1* patterning defects in somite 5 and 6 had apparently corrected by (gray bars, E9.5 Figure 1—figure supplement 1). Hence, these somites partially corrected or regulated their pattern and may account for the relatively low level of rostral vertebral defects at E18.5 (Figure 1E). This correction of rostral somite patterning is found in embryos lacking *Hes7* expression ^40^, which, as we show below, is reduced in *Fgf4* mutants.”

6) One key remaining question is the mechanism of Fgf4 function. I think that the evidence to claim " that FGF4 controls Notch pathway oscillations through the transcriptional repressor, HES7" (Abstract) is not convincing. While indeed Hes7 expression appears irregular in Fgf4 mutants, this could also be interpreted as an indirect consequence of Fgf4 depletion, for instance as consequence of synchronization issues (see above). The finding that Fgf4 and Hes7 genetically synergize does not provide evidence that Hes7 downregulation is a direct consequence of Fgf4 depletion, either. In this regard, it is of note that Hes7 gene expression regulation has been intensively studied and it was shown that the Hes7 enhancer is controlled (and bound) by Tbx6, Mesogenin and Nicd (Hayashi et al., 2018). What is the evidence Fgf4 and FGF signaling controls Hes7 expression directly? I suggest this uncertainty in interpretation is better reflected in the phrasing of the conclusion (again, see Title) and interpretations until the mechanism of Fgf4 is revealed.

One of our important observations is that *Hes7* expression not only “appears irregular” but is reduced by 50%. Such a reduction is likely to cause the observed defects given that *Hes7* heterozygosity or hypomorphic defects, all cause vertebral defects (Sparrow et al., 2012; Stauber et al., 2009; Fujimuro et al., 2014). We contend that the genetic synergy between *Hes7* and *Fgf4* supports this idea; although we agree with the reviewer that it does not prove that *Hes7* is a direct FGF4 target.

In our first draft, we never stated that *Hes7* is a direct FGF4 target. But we can see how a reader can get this impression, given our emphasis on *Hes7*. To correct this, we have edited the Discussion in several places to qualify our *Hes7* model as speculative, Also we

added a new paragraph, incorporating the reviewer’s ideas:

“We propose that this reduction in *Hes7* levels is the primary defect in *Fgf4* mutants, and subsequently causes an irregular activated Notch (NICD) pattern, misexpression of *Mesp2*, and ultimately leads to vertebral defects. A reduction in *Hes7* levels, due to heterozygosity or hypomorphic mutations, has been shown to cause vertebral defects ^25,62,63^. The concept that *Hes7* is generally regulated by FGF signaling is supported by numerous studies ^28,32,64^. Future work will determine whether *Hes7* is a direct FGF4 signaling target, possibly through putative ETS binding sites in the *Hes7* promoter ^65^. Alternatively, FGF4 may indirectly regulate *Hes7* transcription by affecting the activity of TBX6, NICD or MESOGENIN, all shown to regulate *Hes7* expression in the PSM ^65,66^. If so, we speculate such regulation would be post-transcriptional because we find no significant difference in levels of the genes encoding these factors. Additionally, it is possible that FGF4 signaling is interfacing with WNT signaling, which also has been shown to regulate *Hes7*^6529,30^.”